# Multi-functional photonic crystals of modular nanosheets

Seiya Yui[1], Takumi Mihara[1], Tomoki Nishimura [1], Yasuo Ebina [2], Takayoshi Sasaki [2] & Koki Sano [1] ✉

Photonic crystals with periodically ordered nanoscale building blocks can exhibit structural colors, offering a promising optical platform. Among various building blocks, colloidal nanosheets have attracted increasing attention owing to their intrinsic two-dimensionality and stimuli-responsiveness. However, integrating multiple functionalities into nanosheet-based photonic crystals remains challenging due to the structural and colloidal requirements of the nanosheets. Here, we established a universal modular strategy for synthesizing functional hybrid nanosheets and subsequently constructed multi-functional photonic crystals via their self-assembly. By electrostatically integrating negatively charged titanate nanosheets with positively charged nanoparticles, including gold nanoparticles, gold nanorods, and fluorescent silica nanoparticles, we successfully synthesized functional hybrid nanosheets. The enhancement of electrostatic repulsion between these nanosheets enabled the formation of multi-functional photonic crystals with modularly integrated structural color, plasmonic absorption, and fluorescence. Finally, we demonstrated three-dimensional visualization of the photonic nanostructures using confocal microscopy and reversible modulation of the optical properties using magnetic fields and light. This work provides a versatile platform for designing next-generation smart photonic materials with integrated functionalities.

Photonic crystals, composed of long-range ordered nanostructures with periodicities on the scale of several hundred nanometers, exhibit structural colors by selectively reflecting specific wavelengths of light based on Bragg's law[1-4]. Unlike conventional pigments and dyes that rely on light absorption, structural colors offer distinct advantages, including color tunability, long-term stability, and environmental compatibility. Due to these advantages, structural colors are widely observed in nature[5-8], and artificial photonic crystals have been extensively developed through the self-assembly of nanoscale building blocks[3,4] for a wide range of applications, such as sensors[9,10], displays[11,12], printable inks[13,14], photonic pigments[15,16], optical anticounterfeiting[17], and biomedical applications[18]. Among various building blocks, inorganic colloidal nanosheets, synthesized via exfoliation of layered crystals, have emerged as a promising platform for constructing dynamic photonic crystals owing to their intrinsic two-dimensionality and stimuli-responsiveness[19-46]. For example, we recently established a rational strategy for constructing nanosheet-based dynamic photonic crystals by maximizing electrostatic repulsion between nanosheets[19,23]. This strategy enabled the stimuli-responsive modulation of structural colors[19,24,25] and led to the development of unique photonic systems, including mechano-responsive photonic hydrogels[20], dynamic photonic nanostructures capable of mass transport via propagating waves of the collectively movable nanosheets[21], and reconfigurable photonic crystals that can reversibly switch between single and double structural colors[22]. If functional nanosheets could be harnessed to construct dynamic photonic

[1]Department of Chemistry and Materials, Faculty of Textile Science and Technology, Shinshu University 3-15-1 Tokida, Ueda, Nagano, Japan. [2]Research Center for Materials Nanoarchitectonics (MANA), National Institute for Materials Science (NIMS) 1-1 Namiki, Tsukuba, Ibaraki, Japan. ✉e-mail: koki_sano@shinshu-u.ac.jp

crystals, they would offer a new platform for integrating additional functionalities, thereby enabling the development of multi-functional photonic crystals. However, to self-assemble into photonic nanostructures, the nanosheets must meet several stringent requirements: uniform thickness and a high aspect ratio to ensure proper structural ordering, a large surface charge density for strong electrostatic repulsion, and robust structural and colloidal stability for solution processing. Consequently, inorganic nanosheets available for constructing photonic crystals have been restricted to specific compositions, such as titanate[19–22], graphene oxide[23–32], antimony phosphate[33–35], zirconium phosphate[36–39], titanium phosphate[40], niobate[41,42], and clay minerals[43–46]. Although various functionalized inorganic nanosheets (e.g., plasmonic[47–52] and fluorescent[53–59] properties) have been developed for broad applications, for example, by attaching functional nanoparticles onto the nanosheets[47–53], their use in constructing photonic crystals has yet to be realized, primarily due to their difficulty in satisfying the above stringent structural and colloidal criteria. In this context, it remains a significant challenge to develop a universal strategy for synthesizing inorganic nanosheets that not only satisfy the criteria but also possess additional functionalities for constructing multi-functional photonic crystals.

To address this challenge, in this work, we propose a general method to post-functionalize the base nanosheets known to form photonic crystals, while retaining their photonic ability, through surface modification with functional nanoparticles (Fig. 1a). This approach allows for the modular integration of diverse functionalities into nanosheets simply by varying the nanoparticles, thereby enabling the creation of multi-functional photonic crystals with tunable optical properties. As the base nanosheet, we selected titanate nanosheets (TiNSs)[60–62], since we had previously confirmed their ability to form stable photonic nanostructures[19–22]. By electrostatically combining negatively charged TiNSs with a variety of positively charged functional nanoparticles, such as gold nanoparticles (AuNPs)[63], gold nanorods (AuNRs)[64], and fluorescent silica nanoparticles (FSNPs)[65], under optimized conditions, we successfully synthesized structurally and colloidally stable hybrid nanosheets with customizable optical properties (Fig. 1b). Subsequently, by enhancing electrostatic repulsion between these hybrid nanosheets, we expanded their interlayer distance to several hundred nanometers, resulting in multi-functional photonic crystals with modularly integrated structural color, plasmonic absorption, and fluorescence (Fig. 1c). Notably, the use of fluorescent nanosheets enabled direct three-dimensional (3D) visualization of individual nanosheets within the photonic nanostructure using confocal laser scanning microscopy (CLSM). Finally, we demonstrated reversible modulation of the optical properties of the photonic crystals by adjusting the nanosheet orientation through the application of a strong magnetic field (Fig. 1d), as well as reversible tuning of structural color by manipulating the interlayer distance via light irradiation (Fig. 1e), reminiscent of photo-responsive structural colors of marine organisms[6–8]. This work highlights a universal modular strategy for the synthesis of functional nanosheets and the subsequent development of multi-functional photonic crystals through their self-assembly, thereby expanding the design paradigm for next-generation photonic materials with emergent and integrated optical functionalities.

## Results

### Synthesis and characterization of hybrid nanosheets

In this study, we employed negatively charged TiNSs[60–62] with a thickness of 0.75 nm and a lateral size of several micrometers as the base nanosheet. TiNSs are well dispersed in water due to their high surface charge density, forming a periodic nanostructure governed by the balance between electrostatic repulsion and van der Waals attraction, as described by the Derjaguin–Landau–Verwey–Overbeek (DLVO) theory[19,66]. As we previously reported[19–22], the enhancement of the electrostatic repulsion between TiNSs through deionization can increase the interlayer distance between TiNSs up to several hundred nanometers, enabling the formation of photonic crystals with vivid structural colors.

To modularly impart additional functionalities to TiNSs without compromising their ability to form photonic crystals, we electrostatically attached positively charged functional nanoparticles to the negatively charged TiNS surfaces under controlled conditions. As candidates for functional nanoparticles, we selected AuNPs[63] and AuNRs[64] for their plasmonic properties and FSNPs[65] for their fluorescent properties. First, we prepared positively charged functional nanoparticles, where the zeta potentials were measured to be +52 mV for AuNPs, +55 mV for AuNRs, and +21 mV for FSNPs (Supplementary Fig. 1). Transmission electron microscopy (TEM) images and optical characterizations (extinction and/or fluorescence spectra) revealed the structural and optical features of the nanoparticles: AuNPs (diameter: ~17 nm; plasmonic absorption peak: 525 nm; Fig. 2b), AuNRs (short axis: ~21 nm, long axis: ~38 nm; plasmonic absorption peaks: 520 and 615 nm; Fig. 2d), and FSNPs (diameter: ~30 nm; fluorescence peak: 578 nm; Fig. 2f).

To attach the nanoparticles to the surfaces of TiNSs (Fig. 2a and Supplementary Fig. 2a), the aqueous dispersions of nanoparticles were slowly added to a dilute dispersion of TiNSs (zeta potential: −51 mV; Supplementary Fig. 1), affording hybrid nanosheets with functional nanoparticles via electrostatic attraction (AuNP-TiNSs, AuNR-TiNSs, and FSNP-TiNSs). TEM images and extinction and/or fluorescence spectra (AuNP-TiNSs in Fig. 2c and Supplementary Fig. 2b; AuNR-TiNSs in Fig. 2e and Supplementary Fig. 2c; FSNP-TiNSs in Fig. 2g and Supplementary Fig. 2d) confirmed successful attachment of the nanoparticles to the TiNS surfaces while retaining their original optical properties. Importantly, the negative charge of the hybrid nanosheets was almost maintained even after the attachment of positively charged nanoparticles on the surfaces owing to the controlled low concentrations of the added nanoparticles during the synthesis of the hybrid nanosheets, as confirmed by zeta potential measurements (Supplementary Fig. 1), ensuring the large electrostatic repulsion between the nanosheets. Consequently, the dispersions of the hybrid nanosheets exhibited liquid-crystalline behavior similar to that of the original TiNS dispersion, as shown in polarized optical microscopy (POM) images (Supplementary Fig. 3). POM images and extinction or fluorescence spectra of the hybrid nanosheets, taken after 7 days of storage or after heating at 70 °C for 30 min (AuNP-TiNSs in Supplementary Fig. 4; AuNR-TiNSs in Supplementary Fig. 5; FSNP-TiNSs in Supplementary Fig. 6), confirmed that their liquid-crystalline behavior and optical properties remained essentially unchanged, supporting their long-term and thermal stability.

When the concentrations of the added nanoparticles were much higher than the optimized values during the synthesis of the hybrid nanosheets, the resultant nanosheets aggregated and hardly exhibited liquid-crystalline behavior, possibly due to a reduction of the surface charge of the original TiNSs (Supplementary Fig. 7). In contrast, the addition of a smaller amount of functional nanoparticles resulted in colloidally stable hybrid nanosheets that retained liquid-crystalline behavior but showed attenuated optical features (Supplementary Fig. 8). By using a mixture of AuNRs and FSNPs for the synthesis of hybrid nanosheets, we obtained dual-functional hybrid nanosheets with AuNRs and FSNPs (AuNR/FSNP-TiNSs) that exhibited both plasmonic and fluorescent properties (Fig. 2i and Supplementary Figs. 2e and 3e). Even when the ratio between AuNRs and FSNPs was varied while keeping the amount of added AuNRs constant, the plasmonic properties remained essentially unchanged, indicating no significant competitive adsorption between AuNRs and FSNPs (Supplementary Fig. 9).

Remarkably, we successfully visualized individual hybrid nanosheets (FSNP-TiNSs) dispersed in water using fluorescence microscopy (Fig. 2h). It is noteworthy that the real-time movement of

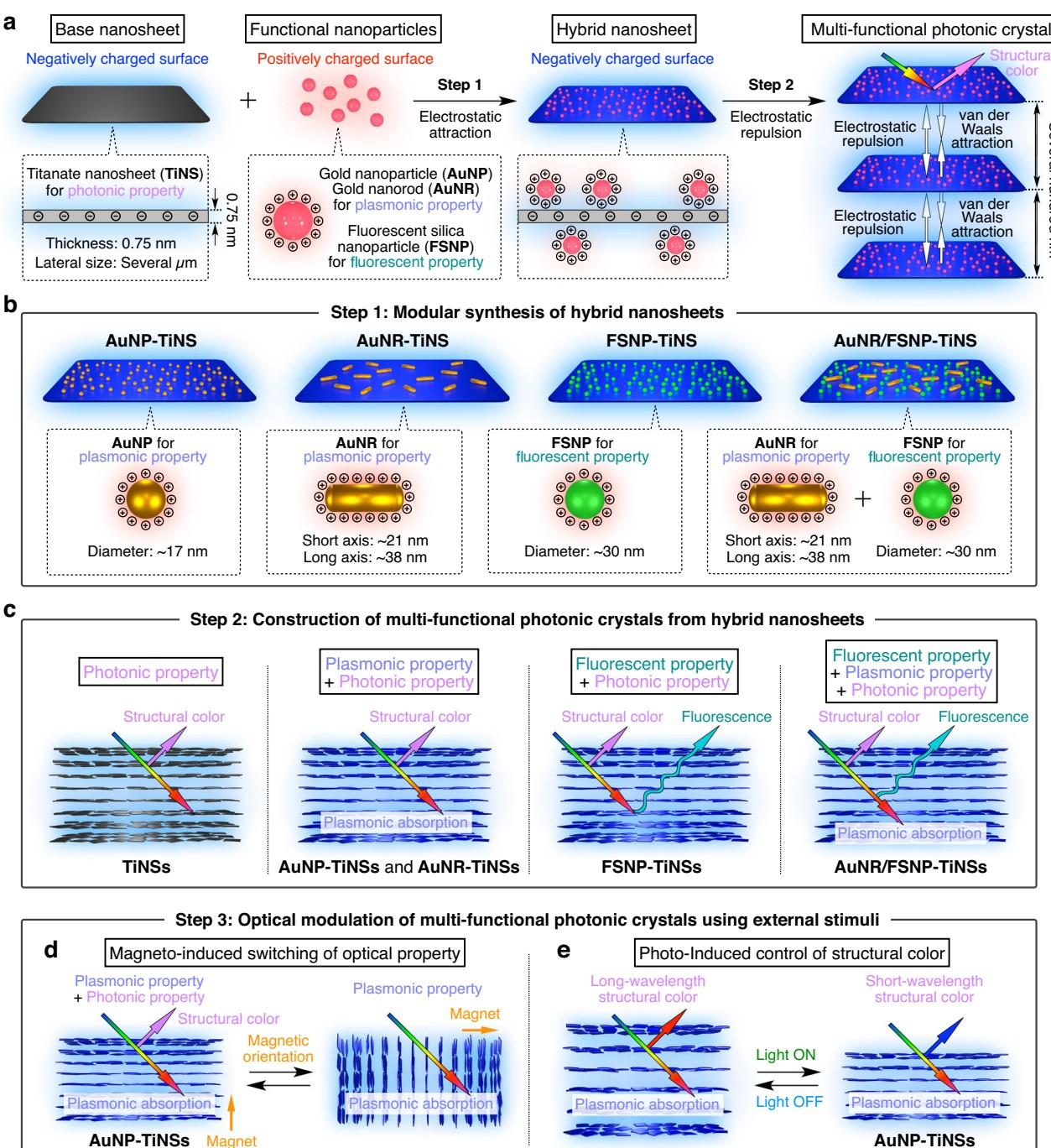

**Fig. 1 | Synthesis and control of multi-functional photonic crystals composed of hybrid nanosheets with modularly integrated properties. a** Schematic illustrations of a modular method for synthesizing functional hybrid nanosheets by electrostatically combining negatively charged titanate nanosheets (TiNSs) with various positively charged functional nanoparticles, including gold nanoparticles (AuNPs), gold nanorods (AuNRs), and fluorescent silica nanoparticles (FSNPs), and constructing multi-functional photonic crystals by enhancing the electrostatic repulsion between the nanosheets. **b** Schematic illustrations of hybrid nanosheets with modularly integrated properties. **c** Schematic illustrations of multi-functional photonic crystals composed of hybrid nanosheets with customizable functionalities, including structural color, plasmonic absorption, and fluorescence. **d, e** Schematic illustrations of (**d**) magneto-induced switching of optical properties and (**e**) photo-induced control of structural color of a photonic crystal of AuNP-TiNSs.

the ultrathin nanosheets can be directly monitored (Supplementary Movie 1), contributing to the analysis of their self-assembly and dynamic behavior. Even after one month of storage (Supplementary Fig. 10a) or heating at 70 °C for 30 min (Supplementary Fig. 10b), the hybrid nanosheets remained clearly visible, further supporting the stable and robust attachment of nanoparticles to the TiNS surfaces in water. These results demonstrate the universality and robustness of our strategy for synthesizing structurally and colloidally

stable hybrid nanosheets with modularly integrated functionalities, offering a versatile platform for the development of multi-functional photonic crystals as well as a wide variety of other smart soft materials.

## Magnetically induced orientation control of hybrid nanosheets

A magnetic field serves as an effective external stimulus to control the orientation of nanosheets. In our previous studies, we demonstrated

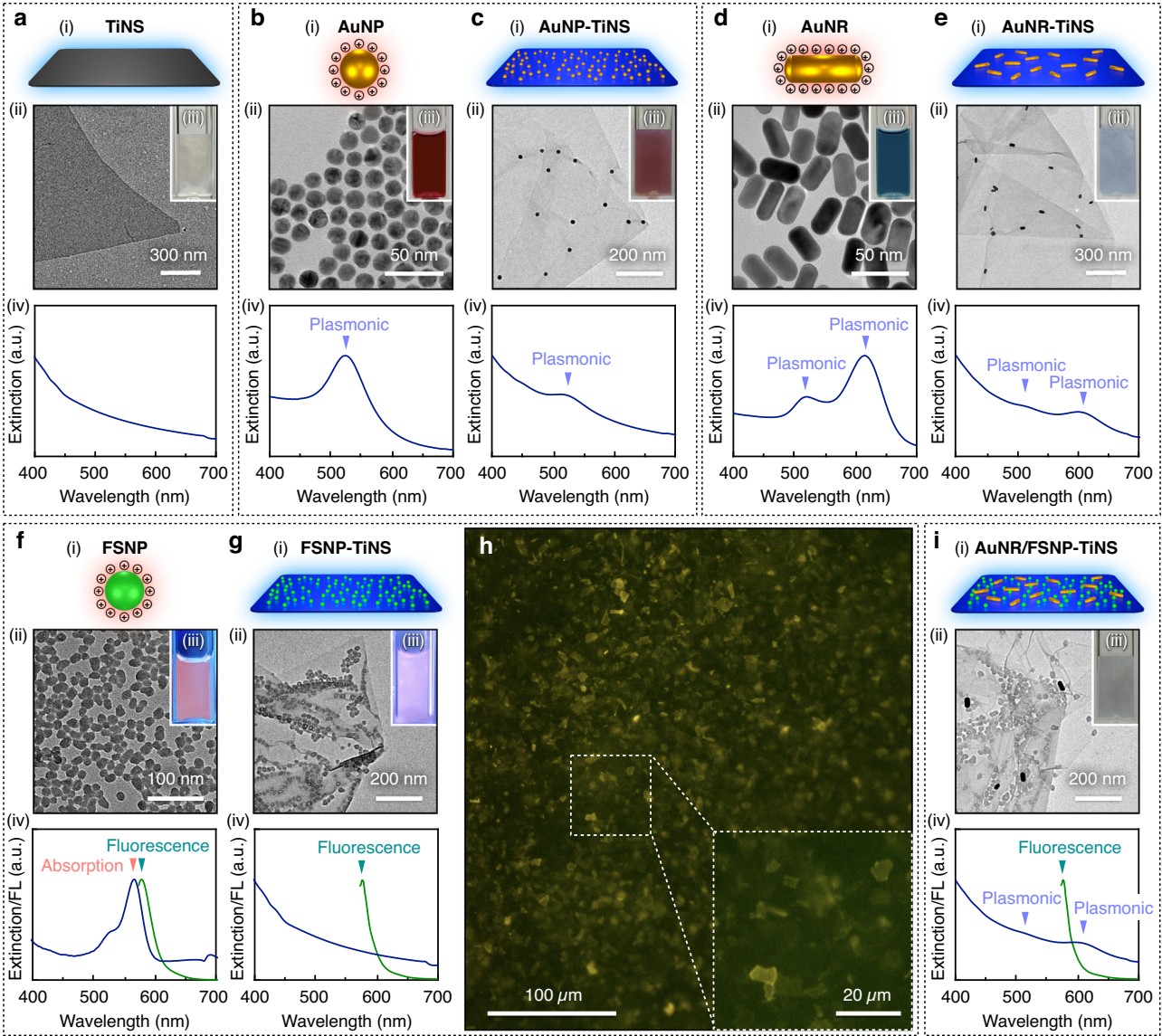

**Fig. 2 | Characteristics of hybrid nanosheets. a–e** (i) Schematic illustrations, (ii) transmission electron microscopy (TEM) images, (iii) optical images of the dispersions, and (iv) extinction spectra of (**a**) titanate nanosheets (TiNSs), (**b**) gold nanoparticles (AuNPs), (**c**) AuNP-functionalized TiNSs (AuNP-TiNSs), (**d**) gold nanorods (AuNRs), and (**e**) AuNR-functionalized TiNSs (AuNR-TiNSs). **f, g** (i) Schematic illustrations, (ii) TEM images, (iii) optical images of the dispersions under UV illumination, and (iv) extinction (navy line) and fluorescence (green line) spectra of (**f**) fluorescent silica nanoparticles (FSNPs) and (**g**) FSNP-functionalized TiNSs (FSNP-TiNSs). **h** Fluorescence microscopy image of an aqueous dispersion of FSNP-TiNSs. **i** (i) Schematic illustration, (ii) TEM image, (iii) optical image of the dispersion, and (iv) extinction (navy line) and fluorescence (green line) spectra of TiNSs functionalized with both AuNRs and FSNPs (AuNR/FSNP-TiNSs).

remote control of the orientation of TiNSs[19–22] and graphene oxide nanosheets[24,25] in their photonic crystals using a strong magnetic field (e.g., 10 T) for the modulation of structural colors. To investigate the magnetically responsive behaviors of the hybrid nanosheets in this work, we performed small-angle X-ray scattering (SAXS) measurements at the SPring-8 synchrotron radiation facility. The 2D-SAXS images of TiNSs (Fig. 3b) suggest that TiNSs were randomly oriented without a magnetic field (Fig. 3a, i), whereas the application of a 12 T magnetic field induced the perpendicular orientation of the TiNS planes to the applied magnetic field (Fig. 3a, ii). The 2D-SAXS images of the hybrid nanosheets exhibit profiles that were nearly identical to those of TiNSs (AuNP-TiNSs in Fig. 3c; AuNR-TiNSs in Fig. 3d; FSNP-TiNSs in Fig. 3e; AuNR/FSNP-TiNSs in Fig. 3f), indicating that the nanosheet planes likewise aligned perpendicular to the magnetic field. The macroscopic orientability was further supported by POM observations (Supplementary Fig. 11). These results revealed that the magnetic orientability of TiNSs is preserved even after the integration of

functional nanoparticles, allowing for precise magnetic control over the orientation of the hybrid nanosheets.

## Construction of multi-functional photonic crystals from hybrid nanosheets

As we previously reported[19–25], the key to constructing dynamic photonic crystals from colloidal nanosheets lies in enhancing the electrostatic repulsion between the nanosheets, for instance, by deionization through repeated centrifugation and redispersion cycles. By applying this approach to AuNP-TiNSs, we successfully constructed a photonic crystal composed of AuNP-TiNSs that exhibited a structural color. Comparison of the SAXS profile and the UV-Vis spectrum of the photonic crystal indicated that the structural color followed Bragg's law (Supplementary Fig. 12)[67]. The UV-Vis spectrum of the AuNP-TiNS dispersion at a concentration of 0.40 wt% in a quartz cuvette (40 × 10 × 1 mm), measured in transmission mode, displayed two characteristic peaks (Fig. 4a): one peak at around 650 nm corresponding to

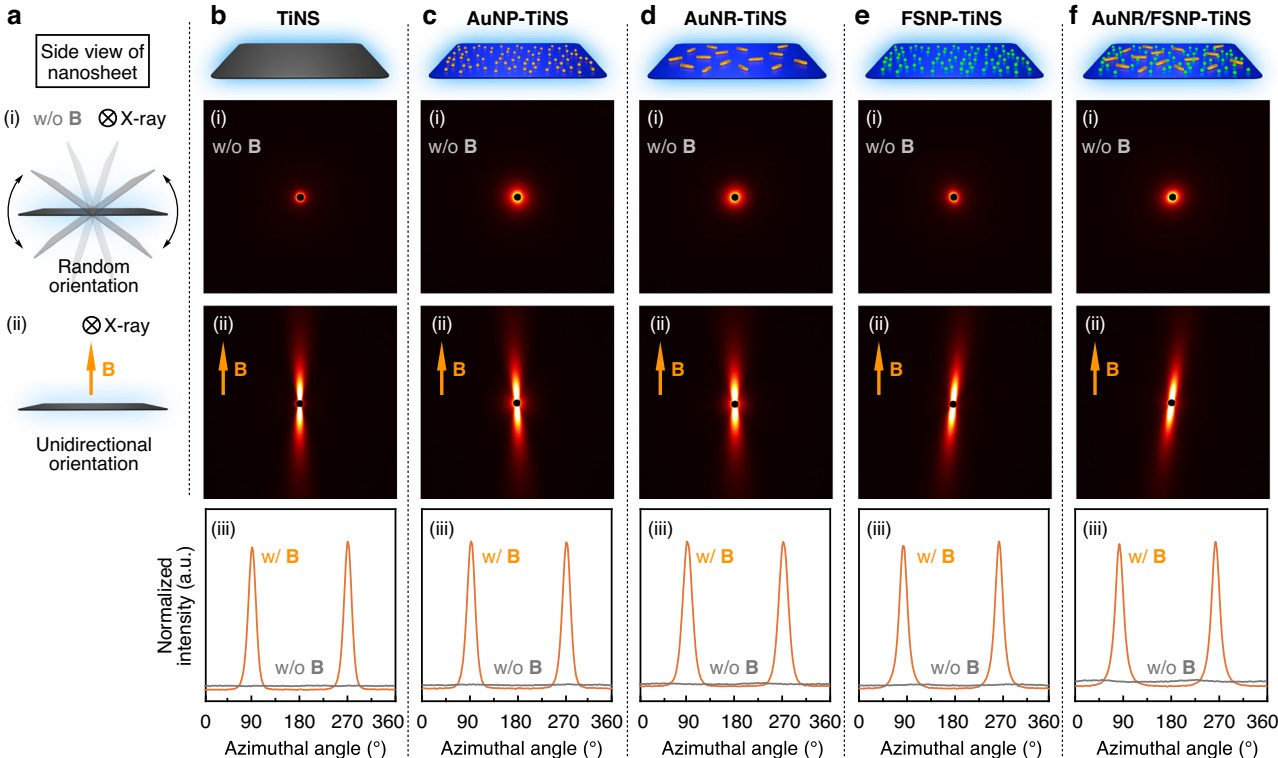

**Fig. 3 | Magnetic orientability of hybrid nanosheets. a** Schematic illustrations of the side view of a nanosheet: (i) random and (ii) perpendicular orientation of the nanosheet plane without and with the application of a magnetic field, respectively. **b–f** Two-dimensional small-angle X-ray scattering (2D-SAXS) profiles of the nanosheets at a concentration of 0.050 wt% (**b**: TiNSs; **c**: AuNP-TiNSs; **d**: AuNR-TiNSs; **e**: FSNP-TiNSs; **f**: AuNR/FSNP-TiNSs) fixed in hydrogels (i) without (0 T) and (ii) with (12 T) the application of a magnetic field. (iii) Azimuthal angle plots obtained from the corresponding 2D-SAXS profiles.

the structural color arising from the photonic nanostructure of the nanosheets and the other peak at 515 nm due to plasmonic absorption of AuNPs on the nanosheet surfaces. Upon applying a 12 T magnetic field parallel to the observation direction (i.e., along the $z$-axis in Fig. 4b), all AuNP-TiNSs aligned perpendicular to the applied magnetic field, resulting in a uniform structural color (Fig. 4b, ii). Consequently, the peak intensity of the structural color increased in the UV-Vis spectrum, while the plasmonic absorption peak remained almost unchanged (Fig. 4b, iii and Supplementary Fig. 13b, i). In the reflection spectra, the reflection peaks corresponding to the structural color were also observed, whereas the plasmonic absorption peak was not detected (Supplementary Fig. 13b, iii). The peak positions of the structural color were almost identical to those of a photonic crystal of pristine TiNSs, although the peaks of AuNP-TiNSs were broader, possibly due to slight structural disorder in the photonic nanostructures (Supplementary Fig. 13a, b). The photonic crystal retained its original optical properties even after 7 days of storage (Supplementary Fig. 14). When the 12 T magnetic field was applied perpendicular to the observation direction (i.e., along the $y$-axis in Fig. 4c), the structural color disappeared, and only the red plasmonic color remained visible (Fig. 4c). Accordingly, the structural color peak disappeared in the UV-Vis spectrum, while the plasmonic absorption peak remained almost constant. These results highlight the potential for magneto-induced modulation of the optical properties in the photonic crystals.

Based on these results, we further constructed multi-functional photonic crystals using other hybrid nanosheets at a concentration of 0.50 wt% and subjected them to the magnetic treatment (AuNR-TiNSs in Fig. 4d; FSNP-TiNSs in Fig. 4e; AuNR/FSNP-TiNSs in Fig. 4f). As expected, a photonic crystal of AuNR-TiNSs exhibited both photonic and plasmonic properties (Fig. 4d). In the UV-Vis spectrum measured in transmission mode, the overlapping peak of the structural color and plasmonic absorption of AuNRs was observed at around

520 nm and the other plasmonic absorption peak of AuNRs was detected at around 620 nm (Fig. 4d and Supplementary Fig. 13d, i). In the reflection spectrum, the reflection peaks of the structural color were similarly observed, whereas the plasmonic absorption peaks were not detected (Supplementary Fig. 13d, iii). The peak positions of the structural color were almost identical to those of a photonic crystal of pristine TiNSs, although the peaks of AuNR-TiNSs were broader, possibly due to slight structural disorder in the photonic nanostructures (Supplementary Fig. 13c, d). Additionally, a photonic crystal composed of FSNP-TiNSs displayed both photonic and fluorescent properties, as evidenced by the structural color peak at 511 nm in the UV-Vis spectrum and the fluorescence peak at 578 nm arising from FSNPs on the nanosheet surfaces in the fluorescence spectrum (Fig. 4e). Finally, we successfully constructed a photonic crystal composed of AuNR/FSNP-TiNSs, which simultaneously exhibited photonic, plasmonic, and fluorescent properties. In this case, the UV-Vis spectrum revealed the overlapping peak corresponding to both the structural color and plasmonic absorption, as well as the other plasmonic absorption peak of AuNRs, while the fluorescence spectrum revealed the emission peak (Fig. 4f).

### Three-dimensional structural characterization of multi-functional photonic crystals

To investigate the internal nanostructures of these photonic crystals, we employed CLSM to three-dimensionally visualize the fluorescent nanosheets (FSNP-TiNSs). After fixing the magnetically treated photonic crystal of FSNP-TiNSs (0.40 wt%) via in situ photo-induced radical polymerization, we performed CLSM imaging of the resultant hydrogel in a wet state. As shown in Fig. 4g, we achieved 3D visualization of the photonic nanostructure composed of nanosheets, in which the individual nanosheets, along with their lamellar arrangement and orientation, were clearly observed despite their ultrathin nature.

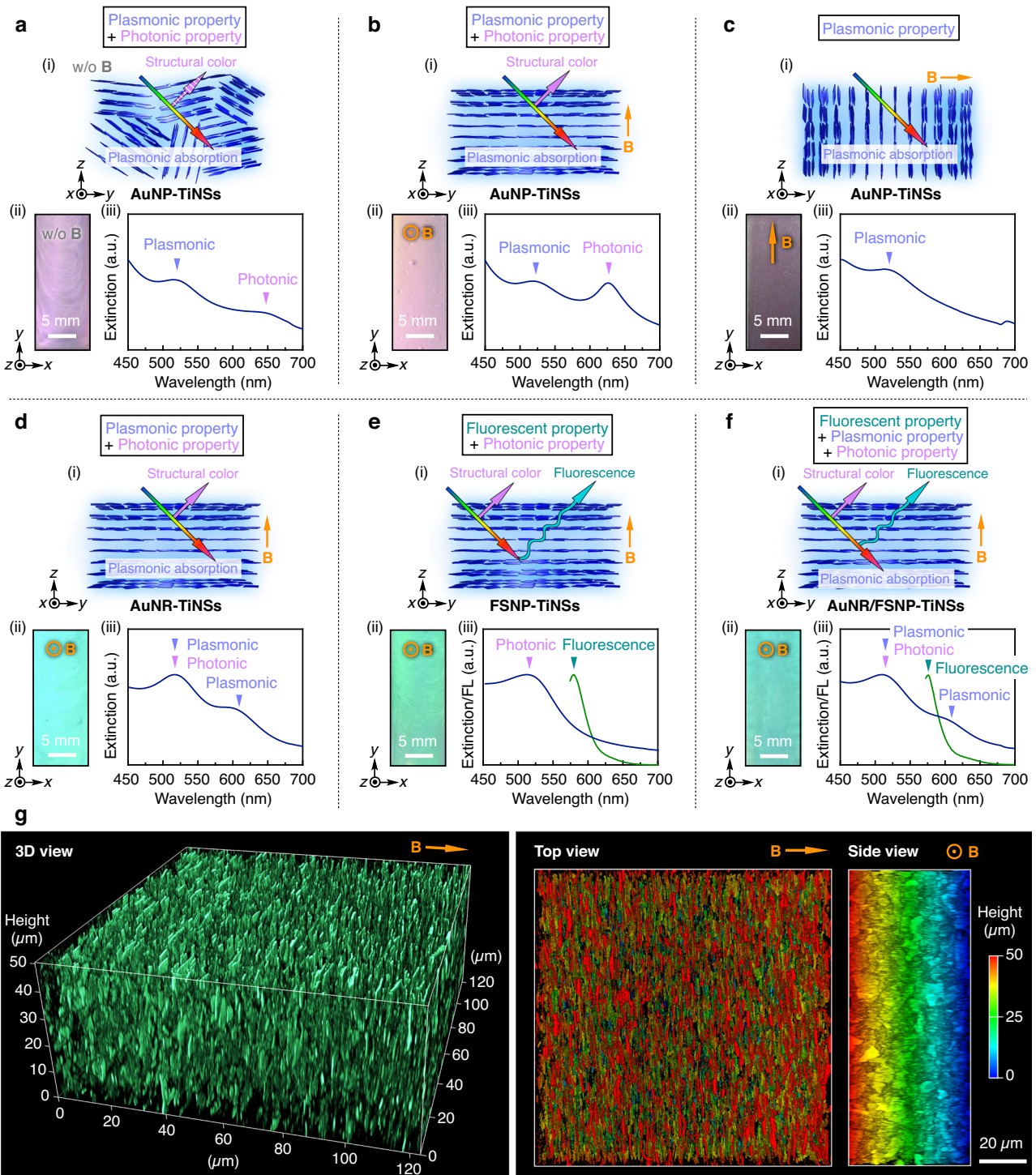

**Fig. 4 | Characteristics of multi-functional photonic crystals composed of hybrid nanosheets. a**–**c** (i) Schematic illustrations, (ii) optical images, and (iii) UV-Vis spectra of the photonic crystal of AuNP-TiNSs (0.40 wt%) (**a**) before and (**b**, **c**) after magnetic application along the (**b**) z-axis and (**c**) y-axis. **d**–**f** (i) Schematic illustrations, (ii) optical images, and (iii) UV-Vis (navy line) and fluorescence (green line) spectra of the magnetically treated photonic crystals of (**d**) AuNR-TiNSs (0.50 wt%), (**e**) FSNP-TiNSs (0.50 wt%), and (**f**) AuNR/FSNP-TiNSs (0.50 wt%). **g** A reconstructed 3D confocal laser scanning microscopy (CLSM) image of the magnetically treated photonic crystal of FSNP-TiNSs (0.40 wt%; left) and the corresponding top and side views (right) obtained using a 570-nm laser.

Moreover, we succeeded in visualizing individual nanosheets dispersed in water without any structural fixation such as hydrogelation. Consequently, we observed the time-dependent structural relaxation of the nanosheets from a magnetically oriented state to a random state (Supplementary Fig. 15 and Supplementary Movie 2) and their dynamic behavior within giant vesicles (Supplementary Fig. 16 and

Supplementary Movie 3). These results are particularly noteworthy given the intrinsic trade-off that exists between the spatial resolution required to visualize individual nanosheets and the ability to resolve their self-assembled nanostructures. Under dilute conditions, isolated fluorescent nanosheets can be visualized using fluorescence microscopy and CLSM[54,55]. However, in the self-assembled or stacked state,

individual nanosheets are generally difficult to resolve due to overlapping and structural complexity, thereby limiting visualization to their macroscopic architecture[56–59]. Consequently, direct visualization of individual nanosheets within their self-assembled nanostructures has remained a significant challenge. In this study, we demonstrate that CLSM imaging with our fluorescent nanosheets overcomes this challenge, because the photonic nanostructure of the nanosheets ensures a sufficiently large interlayer distance, enabling their individual resolution under CLSM. These findings provide a versatile platform for precise analysis and further exploration of the self-assembly and dynamic behavior of colloidal nanosheets.

## Optical control of multi-functional photonic crystals

To investigate the optical controllability of multi-functional photonic crystals, we first examined the effect of nanosheet concentration ([AuNP-TiNS]) on their optical properties, including structural color and plasmonic absorption. An aqueous dispersion of AuNPs alone displayed a plasmonic absorption peak at 525 nm, resulting in a red appearance (Fig. 5a). Upon increasing [AuNP-TiNS] from 0.40 to 0.60 wt%, the first-order structural color peak of the AuNP-TiNS photonic crystals showed a continuous blue shift from 1235 to 877 nm, while the plasmonic absorption peak remained nearly unchanged at around 525 nm. Accordingly, the second-order structural color peak in the visible region was also blue-shifted, accompanied by a gradual color change from pink to yellow, green, and finally blue. This blue shift is attributed to a decrease in the interlayer distance between AuNP-TiNSs with increasing [AuNP-TiNS], in accordance with Bragg's law[19]. We then examined how the ionic concentration ([NaCl]) and pH affected the optical properties. We found that increasing [NaCl] or lowering pH led to a blue shift of the structural color, as the reduced electrostatic repulsion between AuNP-TiNSs decreased their interlayer distance (Supplementary Fig. 17).

Next, we aimed to realize reversible switching of optical properties (i.e., structural colors and plasmonic absorption) by applying a strong magnetic field to align the hybrid nanosheets within the photonic crystal. When a 12 T magnetic field was applied to the photonic crystal of AuNP-TiNSs (0.50 wt%) along the $z$-axis in Fig. 5b, the nanosheet planes aligned perpendicular to the observation direction. As a result, the photonic crystal exhibited a vivid green structural color, and the UV-Vis spectrum displayed two peaks corresponding to the structural color (first order: 1016 nm; second order: 515 nm) and plasmonic absorption (515 nm). In contrast, the application of the magnetic field along the $y$-axis in Fig. 5b induced the nanosheet planes to align parallel to the observation direction. Consequently, the green structural color disappeared and only the red plasmonic color remained visible, where a single peak corresponding to the plasmonic absorption at 513 nm was observed in the UV-Vis spectrum (Fig. 5b). We confirmed angle-dependent changes in the structural color by gradually varying the angle of the applied magnetic field, which showed a blue shift consistent with Bragg's law (Supplementary Fig. 18). The magneto-induced switching was fully reversible, enabling dynamic modulation between photonic/plasmonic and purely plasmonic states. Furthermore, we successfully converted the photonic/plasmonic state into the purely plasmonic state using photonic crystals of AuNP-TiNSs and AuNR-TiNSs with different nanosheet concentrations (Supplementary Fig. 19). Importantly, the photonic crystals of AuNP-TiNSs and AuNR-TiNSs exhibited controllable structural colors as well as plasmonic red and blue colors, respectively, demonstrating their potential as photonic inks with unique reflection- and absorption-based coloration (Supplementary Fig. 20).

It is known that certain marine organisms, including neon tetra fish[6], sapphirinid copepods[7], and brown algae[8], can change their structural colors in response to light. Inspired by such organisms, we envisioned that the photo-thermal effect of AuNPs[68–70] could be harnessed to achieve photo-induced modulation of the structural color in

the AuNP-TiNS photonic crystal. This idea was based on our previous finding that the TiNS-based photonic crystals exhibit thermally responsive structural colors due to temperature-dependent electrostatic repulsion between TiNSs[19,22]. Therefore, we hypothesized that photo-induced heating through the efficient photo-thermal conversion of AuNPs[68–70] could decrease the interlayer distance between AuNP-TiNSs, thereby causing a blue shift in structural color. To test this hypothesis, the photonic crystal of AuNP-TiNSs (0.60 wt%) in a quartz cuvette ($40 \times 10 \times 1$ mm) was irradiated with green LED light (525 nm) corresponding to the plasmonic absorption wavelength (~525 nm), and temperature changes were monitored using a thermal imaging camera. As shown in Fig. 5c, ii and iii, the temperature increased to 43 °C within 10 min under light irradiation and decreased to 28 °C within 10 min after the light was turned off. In contrast, the control photonic crystal composed only of TiNSs (0.60 wt%) exhibited a negligible temperature change under the same conditions (Fig. 5c, iii and Supplementary Fig. 21). The UV-Vis spectra of the photonic crystal of AuNP-TiNSs (0.60 wt%) recorded before and after light irradiation revealed a reversible blue shift in the structural color peak upon 10 min light irradiation and recovery to the original peak position after air cooling (Fig. 5c, iv). These results demonstrate that the photo-thermal effect of integrated AuNPs on the nanosheet surfaces enabled the photo-induced reversible modulation of the structural color, highlighting the potential of multi-functional photonic crystals as a versatile platform for a wide range of applications.

## Discussion

In summary, we have developed a modular strategy for synthesizing hybrid nanosheets with tailored functionalities by electrostatically integrating negatively charged titanate nanosheets, which can inherently self-assemble into photonic crystals, with a wide range of positively charged functional nanoparticles, including fluorescent silica nanoparticles as well as plasmonic gold nanoparticles and nanorods. The resultant hybrid nanosheets retain the intrinsic ability of the original titanate nanosheets to form photonic crystals and exhibit the diverse functionalities imparted by the incorporated nanoparticles. Consequently, we successfully constructed multi-functional photonic crystals with modularly integrated properties, including structural color, plasmonic absorption, and fluorescence, which are essentially different from the conventional multi-functional photonic crystals composed of spherical nanoparticles[71–75]. The modular nature of this strategy facilitates systematic and orthogonal integration of multiple components, offering precise control over the optical properties of the photonic crystals. Moreover, the optical properties can be reversibly modulated by external stimuli, such as magnetic fields and light. Notably, the use of fluorescent nanosheets synthesized via our strategy enabled direct 3D visualization of individual nanosheets within the photonic nanostructure using CLSM. This approach offers a significant advantage for the analysis of the self-assembled nanostructures of colloidal nanosheets, overcoming the limitations of conventional SEM methods, which typically require complicated fixation and drying procedures that often disrupt the original architectures. Overall, these results demonstrate that this modular strategy provides not only a versatile platform for synthesizing functional nanosheets and constructing multi-functional photonic crystals through their self-assembly, but also an effective method for investigating the self-assembled nanostructures and dynamic behavior of nanosheets. We anticipate that this strategy will open new avenues for the design of next-generation photonic materials with emergent and integrated optical functionalities.

## Methods
### General
The magnetic orientation of colloidal nanosheets was carried out using a Cryogenic model CFM-12T-100-H3 superconducting magnet with a

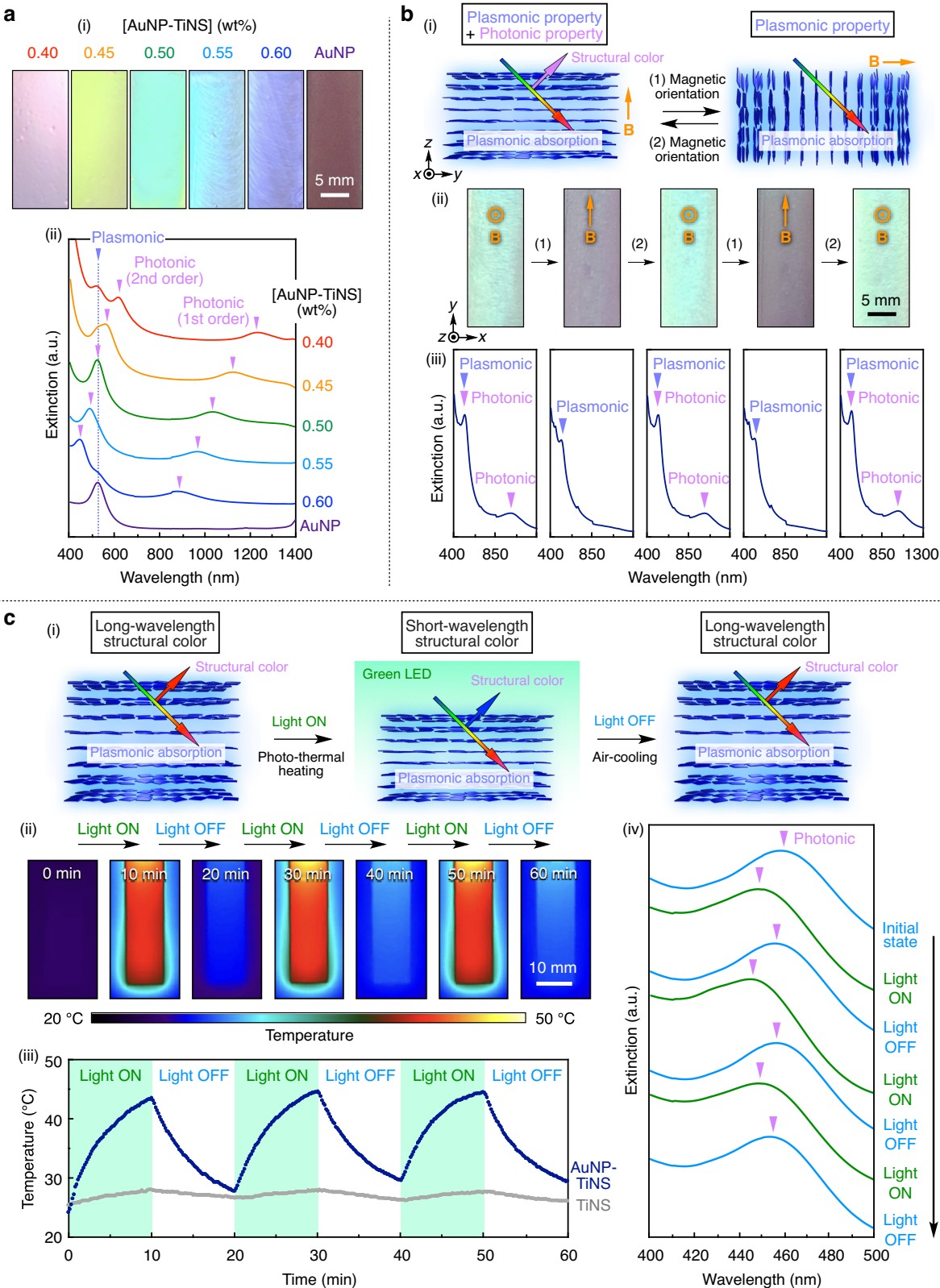

**Fig. 5 | Optical modulation of multi-functional photonic crystals. a** (i) Optical images and (ii) UV-Vis spectra of the magnetically treated photonic crystals of AuNP-TiNSs with different concentrations (0.40–0.60 wt%) and an aqueous dispersion of AuNPs (0.0036 wt%). **b** (i) Schematic illustrations, (ii) optical images, and (iii) UV-Vis spectra of the photonic crystal of AuNP-TiNSs (0.50 wt%) after alternating magnetic applications along the z-axis and y-axis. **c** (i) Schematic illustrations, (ii) thermal images, (iii) corresponding temperature profiles, and (iv) UV-Vis spectra of the magnetically treated photonic crystal of AuNP-TiNSs (0.60 wt%) before and after 10 min of green light irradiation.

100 mm bore. Centrifugation was performed using a TOMY model CAX-571 centrifuge equipped with a TOMY model CA-16 rotor. Transmission electron microscopy (TEM) was carried out using a JEOL model JEM-2100 electron microscope. Polarized optical microscopy (POM) and fluorescence microscopy were performed using a Nikon model Eclipse LV100N POL optical polarizing microscope equipped with a Nikon model LV-UEPI2 universal illuminator and a Nikon model D-LEDI LED light source. Zeta potential measurements were conducted using a Malvern model Zetasizer Pro.

## Materials

Hexadecyltrimethylammonium bromide (CTAB), sodium 3-methylsalicylate, *N,N*-dimethylacrylamide, *N,N'*-methylenebisacrylamide, sodium chloride (NaCl), and 1,2-dioleoyl-*sn*-glycero-3-phosphocholine (DOPC) were purchased from Tokyo Chemical Industry (TCI). Trisodium citrate, hydrochloric acid (HCl), D-glucose, methanol, and dichloromethane were purchased from FUJIFILM Wako Pure Chemical Corporation. Gold(III) chloride trihydrate, silver(I) nitrate, sodium borohydride, ascorbic acid, and 1,2-dioleoyl-*sn*-glycero-3-phosphoethanolamine-*N*-(7-nitro-2-1,3-benzoxadiazol-4-yl) (NBD-PE) were purchased from Sigma-Aldrich. Fluorescent silica nanoparticles (NH$_2$-modified sicastar-redF; diameter: 30 nm) were purchased from micromod. Ultrapure water was produced by a Millipore model Milli-Q IQ 7003 water purification system and used throughout the experiments. Polymerization inhibitors in *N,N*-dimethylacrylamide were removed using inhibitor removers (Sigma-Aldrich) prior to use. The as-received aqueous dispersions (30 and 15 μL) of fluorescent silica nanoparticles (FSNPs; 2.5 wt%) were diluted to final volumes of 1.5 mL and 1.0 mL using HCl solutions (1.0 mM and 1.5 mM, respectively). The resultant dilute FSNP dispersions (0.050 wt% and 0.038 wt%) were used for the synthesis of hybrid nanosheets (FSNP-TiNSs and AuNR/FSNP-TiNSs, respectively). An aqueous dispersion of titanate nanosheets (TiNSs) with tetramethylammonium countercations was prepared according to a reported method[60,61]. After deionization by repeated centrifugation and redispersion cycles[19–22], the TiNS dispersion was used for the synthesis of hybrid nanosheets. Unless otherwise noted, all reagents were used as received from commercial suppliers.

## Synthesis of positively charged gold nanoparticles

Gold nanoparticles (AuNPs) were synthesized according to a reported method[69,70]. Briefly, an aqueous solution of gold(III) chloride trihydrate (0.25 mM, 300 mL) was heated to 90 °C, and an aqueous solution of trisodium citrate (40 mM, 6.0 mL) was rapidly injected into the solution under stirring at 600 rpm. After 10 min of reaction, an aqueous dispersion of negatively charged AuNPs was obtained. After cooling to room temperature, 40 mL of the dispersion was purified by two cycles of centrifugation at 5000×*g* for 1 h and redispersion in water, and finally concentrated to a volume of 5.0 mL. To convert the surface charge of AuNPs from negative to positive, an aqueous solution of CTAB (20 mM, 2.0 mL) was rapidly added to the AuNP dispersion (5.0 mL). The excess CTAB was removed by centrifugation at 5000×*g* for 1 h. After this treatment, the zeta potential of AuNPs changed from −33 mV to +52 mV, confirming the successful modification to positively charged AuNPs.

## Synthesis of positively charged gold nanorods

Gold nanorods (AuNRs) were synthesized according to a reported method[76]. Briefly, to prepare the seed dispersion, an aqueous solution of sodium borohydride (6.0 mM, 1.0 mL) was rapidly injected at 25 °C into 10 mL of an aqueous solution containing gold(III) chloride trihydrate (0.25 mM) and CTAB (100 mM) under vigorous stirring at 1200 rpm. After 2 min of reaction, the stirring was stopped, and the dispersion was left undisturbed for 1 h. To prepare the growth solution, 9.0 g of CTAB and 1.1 g of sodium 3-methylsalicylate were dissolved in

250 mL of hot water, followed by the addition of an aqueous solution of silver(I) nitrate (4.0 mM, 6.0 mL) at 30 °C, and the mixture was kept undisturbed for 15 min. Subsequently, an aqueous solution of gold(III) chloride trihydrate (1.0 mM, 250 mL) was added under stirring at 400 rpm for 15 min, followed by the rapid addition of an aqueous solution of ascorbic acid (64 mM, 1.0 mL) under vigorous stirring for 30 s. Finally, 0.80 mL of the seed dispersion was rapidly added into the growth solution under stirring for 30 s, and the mixture was left undisturbed at 30 °C for at least 12 h, resulting in an aqueous dispersion of positively charged AuNRs. The dispersion was purified by centrifugation at 20,000×*g* for 1 h and redispersion in water, followed by an additional cycle of centrifugation at 5000×*g* for 1 h and redispersion in water.

## Synthesis of hybrid nanosheets and construction of multifunctional photonic crystals

To synthesize AuNP-TiNSs, an aqueous dispersion of positively charged AuNPs (0.0045 wt%, 2.0 mL) was slowly added to an aqueous dispersion of TiNSs (0.050 wt%, 10 mL) under stirring at 500 rpm. To synthesize AuNR-TiNSs, an aqueous dispersion of positively charged AuNRs (0.0053 wt%, 2.0 mL) was slowly added to an aqueous dispersion of TiNSs (0.050 wt%, 10 mL) under stirring at 500 rpm. To synthesize FSNP-TiNSs, an aqueous dispersion of positively charged FSNPs (0.050 wt%, 1.5 mL) was added in a stepwise manner to an aqueous dispersion of TiNSs (0.050 wt%, 10 mL). To synthesize AuNR/FSNP-TiNSs, a mixture of the AuNR dispersion (0.011 wt%, 0.50 mL) and the FSNP dispersion (0.038 wt%, 1.0 mL) was slowly added to an aqueous dispersion of TiNSs (0.050 wt%, 10 mL) under stirring at 500 rpm. The resultant dispersions were subjected to two cycles of centrifugation at 20,000×*g* for 20 min and redispersion in water, leading to the construction of multi-functional photonic crystals.

## Optical characterizations

UV-Vis spectra and fluorescence spectra were recorded using a JASCO model V-770 spectrophotometer and a JASCO model FP-8350 spectrofluorometer, respectively. Extinction spectra ($E = -\log_{10}(I/I_0)$, where $I$ and $I_0$ denote the transmitted intensities through the sample and the reference, respectively) in Fig. 2 and Supplementary Figs. 4, 5, 8, and 9 were acquired at room temperature for aqueous dispersions ([TiNS] = 0.20 wt%; [AuNP] = 0.0036 wt%; [AuNP-TiNS] = 0.20 wt%; [AuNR] = 0.0053 wt%; [AuNR-TiNS] = 0.20 wt%; [FSNP] = 0.083 wt%; [FSNP-TiNS] = 0.20 wt%; [AuNR/FSNP-TiNS] = 0.20 wt%) in 1-mm-thick quartz cuvettes (40 × 10 × 1 mm). Fluorescence spectra (Fig. 2 and Supplementary Fig. 6) were recorded at room temperature with an excitation wavelength of 565 nm for aqueous dispersions ([FSNP] = 0.083 wt%; [FSNP-TiNS] = 0.20 wt%; [AuNR/FSNP-TiNS] = 0.20 wt%) in quartz cuvettes (45 × 10 × 10 mm). All spectra in Fig. 2 were normalized to their maximum values. Optical images shown in Fig. 2 were taken of aqueous dispersions ([TiNS] = 0.20 wt%; [AuNP] = 0.018 wt%; [AuNP-TiNS] = 0.20 wt%; [AuNR] = 0.0053 wt%; [AuNR-TiNS] = 0.20 wt%; [FSNP] = 0.050 wt%; [FSNP-TiNS] = 0.20 wt%; [AuNR/FSNP-TiNS] = 0.20 wt%). Fluorescence microscopy images shown in Fig. 2h and Supplementary Fig. 10 were obtained from aqueous dispersions of FSNP-TiNSs (0.0050 wt%) under illumination at 550 nm. Extinction spectra (transmission mode), fluorescence spectra (excitation wavelength: 565 nm), and optical images (Figs. 4 and 5 and Supplementary Figs. 13, 14, and 17–19) were acquired at room temperature for the multi-functional photonic crystals using 1-mm-thick quartz cuvettes (40 × 10 × 1 mm). All spectra in Fig. 4 were normalized to their maximum values, whereas those in Fig. 5a were normalized to the height of the first-order structural color peak of each spectrum. Reflection spectra (Supplementary Figs. 12b and 13, iii) were recorded at room temperature on a JASCO model V-770 spectrophotometer with a JASCO model ARSN-917 manual absolute reflectance measurement unit at an incidence angle of 5°.

## Small-angle X-ray scattering (SAXS) measurements

The SAXS measurements in Fig. 3 were conducted at the BL40B2 beamline of the SPring-8 synchrotron radiation facility (Hyogo, Japan) using a Dectris model PILATUS3 S 2 M photon-counting detector (X-ray wavelength: 1.0 Å; sample-to-detector distance: 2.1 m). The samples were prepared as follows. First, precursor dispersions were prepared so that the final concentrations of nanosheets (TiNSs, AuNP-TiNSs, AuNR-TiNSs, FSNP-TiNSs, and AuNR/FSNP-TiNSs), $N,N$-dimethylacrylamide, and $N,N'$-methylenebisacrylamide were 0.050, 6.0, and 0.060 wt% in water, respectively. Then, the dispersions were poured into 1-mm-thick containers and placed in the bore of a superconducting magnet. A 12 T magnetic field was applied parallel to the container surface for 30 min. To fix the magnetically oriented nanosheets within hydrogels, in situ photo-polymerization was carried out by UV irradiation for 30 min using an USHIO model OPM2-502H super high-pressure mercury lamp (500 W). The macroscopic orientation of the nanosheets was confirmed by POM observations under crossed Nicols using the resultant hydrogel samples (Supplementary Fig. 11). Samples without the magnetic application were prepared in the same manner, except for the absence of a magnetic field. The resultant samples were subjected to SAXS measurements, with the incident X-ray beam directed perpendicular to the front surface of the samples. The resulting 2D-SAXS images were converted into azimuthal angle plots using FIT2D. The 1D-SAXS profile in Supplementary Fig. 12a was acquired at room temperature for the photonic crystal of AuNP-TiNSs (2.0 wt%) in a quartz capillary (diameter: 2 mm) using a Rigaku model NANOPIX 3.5 m system equipped with a Rigaku model HyPix-6000 detector (X-ray wavelength: 1.54 Å; sample-to-detector distance: 1.4 m).

## Confocal laser scanning microscopy (CLSM) observations

Confocal laser scanning microscopy (CLSM) was carried out using a Leica STELLARIS 8 confocal microscope platform in Lightning mode. The hydrogel sample was prepared as follows. First, the precursor dispersion was prepared so that the final concentrations of FSNP-TiNSs, $N,N$-dimethylacrylamide, and $N,N'$-methylenebisacrylamide were 0.40, 10, and 0.10 wt% in water, respectively. Then, the dispersion was poured into a 0.5-mm-thick container and placed in the bore of a superconducting magnet. A 12 T magnetic field was applied parallel to the container surface for 30 min. To fix the magnetically oriented nanosheets within a hydrogel, in situ photo-polymerization was carried out by UV irradiation for 10 min using an USHIO model OPM2-502H super high-pressure mercury lamp (500 W). The resultant sample was observed at room temperature using CLSM with a 570-nm laser, and 2D images were acquired at a vertical step size of 0.3 μm. These 2D images were reconstructed to provide 3D information regarding the nanosheet orientation within the hydrogel (Fig. 4g). To investigate the structural relaxation, an aqueous dispersion of FSNP-TiNSs (0.40 wt%) was poured into a 1-mm-thick container and magnetically treated at 50 °C for 30 min (12 T). Time-dependent changes in the resultant sample were monitored at 1-min intervals at room temperature using CLSM with a 550-nm laser for 5 h (Supplementary Fig. 15 and Supplementary Movie 2). To visualize the dynamic behavior of nanosheets, the sample was prepared as follows. First, a mixture of a methanol solution of DOPC (10 mM, 7.5 μL), a methanol solution of D-glucose (20 mM, 75 μL), and a dichloromethane solution of NBD-PE (0.11 mM, 0.69 μL) was poured into a 1-mm-thick container and completely dried to remove the solvents. Then, an aqueous dispersion of FSNP-TiNSs (0.40 wt%) was added into the container and left undisturbed for at least 12 h. Time-lapse CLSM was performed at room temperature using a 460-nm laser for giant vesicles and a 550-nm laser for the nanosheets (Supplementary Fig. 16 and Supplementary Movie 3).

## Magnetic orientation of multi-functional photonic crystals

Typically, aqueous dispersions of hybrid nanosheets (AuNP-TiNSs, AuNR-TiNSs, FSNP-TiNSs, and AuNR/FSNP-TiNSs) in a 1-mm-thick quartz cuvette ($40 \times 10 \times 1$ mm) were placed in the bore of a superconducting magnet, and a 12 T magnetic field was applied either perpendicular or parallel to the cuvette surface at 70 °C for 30 min. After air cooling to room temperature and removal of the magnetic field, observations and measurements were carried out (Figs. 4 and 5a and Supplementary Figs. 13, 14, 17–19). In Supplementary Fig. 18, the angle of the magnetic field with respect to the cuvette surface was gradually changed from 90° to 40°. To achieve reversible switching of optical properties (i.e., structural colors and plasmonic absorption) as shown in Fig. 5b, this process was repeated at 50 °C for three cycles, after changing the direction of the magnetic field each time.

## Photo-induced modulation of structural color

AuNP-TiNSs were synthesized by slowly adding an aqueous dispersion of positively charged AuNPs (0.0045 wt%, 3.0 mL) to the TiNS dispersion (0.050 wt%, 10 mL) under stirring at 500 rpm. The resultant dispersion was subjected to two cycles of centrifugation at 20,000×$g$ for 20 min and redispersion in water, resulting in a photonic crystal of AuNP-TiNSs that exhibited a structural color. The photonic crystals of AuNP-TiNSs (0.60 wt%) and TiNSs (0.60 wt%) were poured into 1-mm-thick quartz cuvettes ($40 \times 10 \times 1$ mm) and placed in the bore of a superconducting magnet. A 12 T magnetic field was applied perpendicular to the cuvette surface at 80 °C for 30 min. After air cooling to room temperature and removal of the magnetic field, observations and measurements were carried out (Fig. 5c). The resultant photonic crystals were irradiated with green LED light (a Thorlabs model SOLIS-525C high-power LED light; 525 nm, 3.1 W) for 10 min, followed by air cooling for 10 min after the light was turned off. The photo-induced temperature changes were monitored using time-dependent thermal images recorded with a FLIR T530 thermal imaging camera. The photo-induced changes in the structural color of the AuNP-TiNS photonic crystal (0.60 wt%) were evaluated by UV-Vis spectroscopy before and after each light irradiation.

## Data availability

The data that support the findings of this study are available within the paper and its Supplementary Information. Additional data related to the paper are available from the corresponding authors upon request. Source data are provided with this paper.

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

## Acknowledgements

This work was supported by JST FOREST Program Grant Number JPMJFR223O and JST CREST Grant Number JPMJCR23O1, Japan (K.S.), as well as JSPS KAKENHI Grant Number JP23K23325 (K.S.). K.S. acknowledges the Kurita Water and Environment Foundation (KWEF, Japan). The SAXS measurements were performed at the BL40B2 beamline of SPring-8 under proposal number 2024B1105.

## Author contributions

S.Y. and K.S. designed the experiments. S.Y. performed all experiments, except for the small-angle X-ray scattering measurements at SPring-8, which were conducted by T.N. T.M. optimized the experimental conditions and supported the synthesis of gold nanorods. Y.E. and T.S. synthesized the aqueous dispersion of titanate nanosheets. S.Y. and K.S. analyzed the data and wrote the manuscript. All authors discussed the results and commented on the manuscript. K.S. conceived, designed, and supervised the project.

## Competing interests

The authors declare no competing interests.
