## [Transparent Peer Review file · Nature Communications]

Multi-functional photonic crystals of modular nanosheets

Corresponding Author: Professor Koki Sano

Version 0:

Reviewer comments:

Reviewer #1

(Remarks to the Author)

General Comments:

In this manuscript, the authors present a series of optically responsive structures fabricated by electrostatically integrating negatively charged titanate nanosheets with various positively charged functional nanoparticles. The data are presented in a thorough and technically sound manner. However, the work falls short in demonstrating significant innovation within the field. The current approach appears to be an incremental extension of existing work rather than a transformative advance.

Specific Comments:

1. The literature cited throughout the manuscript is largely outdated, with many references dating back five years or more. Incorporating recent studies would strengthen the relevance and positioning of the work.
2. The reported design largely revolves around nanoparticle doping of titanate nanosheets, but does not convincingly reflect the claimed modularity or novel functionality. Furthermore, the manuscript does not sufficiently address existing material challenges or clarify how this system overcomes them.
3. Although the multifunctionality of the composite material is emphasized, the demonstrated responses to external stimuli remain relatively simple and limited. Given the extensive number of similar reports in recent literature, the current results do not convincingly showcase a significant advancement in functional material design.

Reviewer #2

(Remarks to the Author)

This article focuses on the development of multi-functional 2-D nanosheets and their assembly into 1D liquid photonic crystals (LPCs). The study of these not only deepens our fundamental understanding of anisotropic particle interactions but also enables the rational design of functional materials with tailored optical, mechanical, or electronic properties.

In this paper, TiNS colloidal nanosheets functionalized with gold nanoparticles, gold nanorods, or fluorescent silica nanoparticles were prepared for the self-assembly of multifunctional liquid photonic crystals. The structural color features of liquid photonic crystals are tunable by aligning nanosheets with a magnetic field or through light-induced interlayer spacing changes. Also, the integration of fluorescent particles in the system allows for 3D visualization of individual sheets or the self-assembly processes of photonic structures via confocal microscopy.

The study addresses an interesting topic, but several key aspects require clarification or further development:

(1) As the Bragg equation for 1D PCs ($\lambda = 2nd \cos \theta$) is simple, the authors attribute the observed color changes to variations in interlayer spacing d , and estimate the d according to this formula as well. In their earlier publications, the possible TiNS interlayer distance was also estimated using DLVO theory (Nat. Commun. 7, 12559 (2016)). Since the SAXS experiments have already been carried out, and the resolution of SPring-8 facility is extremely high, it is confusing why they did not provide direct measurement results of interlayer spacing d , through wavevector transfer $q = 4\pi \sin \theta / \lambda$. In addition, by comparing the reflection spectra with SAXS data $q = 4\pi \sin \theta / \lambda$ (see Adv. Mater. 2010, 22, 2871–2880), the effective refractive index n in the Bragg equation ($\lambda = 2nd \cos \theta$) can be calculated, which may provide insight into the electrostatic repulsion between 2D colloids both in the vertical (inter-plane) and the horizontal (in-plane) direction for ordering, or providing practical

packing density of the 2D nanosheets within crystalline LPC domains (which should be higher than in the bulk solution).

(2) Fluorescent nanoparticles are incorporated into the nanosheets, enabling visualization of the dynamic self-assembly of 2d nanosheets. But the current implementation is limited to static 3D visualization of the photonic nanostructure, which does not fully demonstrate the advantages and advancements offered by this technique. Given the successful incorporation of fluorescent nanoparticles or dyes into the nanosheets and their three-dimensional visualization in previous studies, it is recommended to visualize the dynamic colloidal assembly process, such as nucleation and growth of photonic crystal domains; dynamic phase transition from a disordered to ordered liquid crystalline states, or kinetics of alignment under external fields (magnetic, electric, shear), making the findings more impactful.

(3) It is recommended that the authors present absolute reflectance measurement results instead of relative reflectance of 1D LPCs, and their transmittance information is also necessary for understanding of their optical properties.

(4) The influence of ion strength, PH changes on the reflectance spectra of 1D LPCs, and long-term stability of the systems (since the Au and fluorescent parties are static electricity adsorbed) should be investigated.

(5) For self-assembly of spherical, polyhedral, or 2d nanoparticles, the ordered packing of particles are supposed to feature a higher packing density, and thereby a stronger reflectance and larger wavelengths of reflective peaks, when compared with disordered systems. But the peak positions of Fig. 4a and 4b showed the opposite tendency, which should be caused by external magnetic field forces, thereby theoretical fundamentals or experimental results of collective effects of magnetic field and electrostatic repulsion on the self-assembly and optical properties of 1D LPCs should be provided.

Reviewer #3

(Remarks to the Author)

This manuscript titled "Multi-functional photonic crystals of modular nanosheets" presents a significant advancement in the field of photonic crystals. The authors introduce a novel and universal modular strategy to synthesize functional hybrid nanosheets and construct multi-functional photonic crystals with customizable properties through self-assembly. The work is highly innovative as it successfully integrates multiple functionalities into nanosheet-based photonic crystals, which has been a considerable challenge due to the strict structural and colloidal requirements. The strategy not only leverages the unique advantages of colloidal nanosheets but also demonstrates impressive controllability and versatility in optical functionalities, such as structural color, plasmonic absorption, and fluorescence. The logical flow of the manuscript is clear, starting from the synthesis of hybrid nanosheets to the construction and characterization of photonic crystals, providing a comprehensive understanding of the research. I believe that with a minor revision addressing the following concerns, it will be more robust and convincing.

1. Regarding the adsorption of positively charged nanoparticles on negatively charged sheets, the authors should investigate how the amount of added nanoparticles affects the formation of composite structures and their properties. They mention that adding too much can affect dispersibility but fail to provide supporting data. Additionally, the authors need to clarify why maintaining a negative charge is essential for stability and whether a positive charge could achieve the same effect through electrostatic repulsion.

2. The authors emphasize the importance of charge interactions for maintaining photonic crystals. For multi-component systems, it is crucial to consider the local heterogeneity and double-electricity resulting from the adsorption of positively charged particles on negatively charged surfaces, which significantly influence the overall properties. The authors are advised to conduct additional control experiments and simulations to gain deeper insights into the role of charge interactions.

3. The practical applications of the synthesized photonic crystals should be demonstrated. While the crystals exhibit excellent controllability and multi-functionality, the authors do not show their performance in real-world applications. For example, could the rotation of nanosheets be observed within cells?

4. The process of adsorbing nanoparticles onto nanosheets is not novel in itself. The innovation lies in maintaining the characteristics of photonic crystals while forming composite structures. The authors should further discuss this aspect in the Introduction to highlight the novelty of their approach, these discussions can also make the article accessible to a wider range of readers.

Reviewer #4

(Remarks to the Author)

This work proposes an innovative modular strategy to construct multifunctional photonic crystals with structural color, plasma absorption, and fluorescence by combining functional nanoparticles (AuNPs, AuNRs, FSNPs) with titanate nanosheets (TiNSs) through electrostatic attraction. The multifunctional photonic crystals exhibit good color and structural tunability by applying external stimuli (light, magnetic fields, etc.) and may find new applications in optical devices. Overall, this work is interesting and well organized, and therefore I would recommend the publication of this work after proper revision. Here are some comments.

1. How is the optimal concentration of functional nanoparticles and nanosheets determined? Do variations in these concentrations influence the optical properties of the resulting hybrid nanosheets?

2. Is there competitive adsorption between AuNRs and FSNPs on TiNSs surfaces? Is the particle distribution across nanosheets uniform? If not, how does adsorption heterogeneity affect optical performance?

3. Quantitative spectral data after a long-term storage and after heating are suggested to be supplemented to further prove the stability of the prepared nanosheets.
4. The explanation of the magnetic field switching structure color is insufficient. The change of the AuNP-TiNSs spectrum with the change of the magnetic field direction during the process of magnetic field regulating the structure color should be supplemented to prove that the disappearance of color satisfies the Bragg diffraction equation.
5. Why does AuNR/FSNP-TiNSs exhibit a lower zeta potential than pure TiNSs?
6. What are the advantages of using nanosheet compared to nanospheres (Adv. Funct. Mater. 2024, 2411670 and Nature Communications, 2024, 15, 5643) in constructing multifunctional photonic crystals? These works should be discussed in the revision.
7. In Fig. 5b, when a magnetic field with a vertical viewing Angle is applied, it can be observed that the secondary diffraction peaks overlap with the plasma absorption peaks. Why is the intensity at this time higher than that of the plasma absorption peaks when a magnetic field with a parallel viewing Angle is applied?
8. Why are the intensities of the first-order diffraction peaks in Fig.5 all significantly lower than those of the second-order diffraction peaks?
9. The authors are suggested to provide the reflective spectra of the pure Ti nanosheets-based photonic crystals and multifunctional photonic crystals to show the influence of Au nanorods on the photonic bandgaps.

Version 1:

Reviewer comments:

Reviewer #2

(Remarks to the Author)

The authors have carefully addressed all the comments raised during the first review round. The revised manuscript now presents a more comprehensive study on the modular construction of multifunctional photonic crystals.

The additional SAXS measurements, optical characterizations (absolute reflectance and transmittance), and dynamic CLSM visualization experiments significantly strengthen the technical rigor of the work. The newly included discussions on the effects of ionic strength, pH, and long-term stability, as well as the expanded explanation of the charge-interaction mechanism and magnetic/orientation control, effectively resolve the previous concerns.

Overall, the manuscript has been substantially improved in both scientific content and presentation quality. The authors' responses are satisfactory and the revisions have enhanced the clarity, completeness, and impact of the work.

I find the manuscript suitable for publication in its current form.

Recommendation: Accept as it stands.

Reviewer #3

(Remarks to the Author)

The author has answered all the raised questions, I think its good enough for publication without the need for further revision.

Reviewer #4

(Remarks to the Author)

The authors have carefully addressed my issues, and I am satisfied with the revision. Therefore, I recommend the acceptance of this work without further change.

Point-by-point Response to Reviewer #1:

General Comments:

In this manuscript, the authors present a series of optically responsive structures fabricated by electrostatically integrating negatively charged titanate nanosheets with various positively charged functional nanoparticles. The data are presented in a thorough and technically sound manner. However, the work falls short in demonstrating significant innovation within the field. The current approach appears to be an incremental extension of existing work rather than a transformative advance.

=> We sincerely appreciate your thoughtful and thorough review, as well as your constructive comments and suggestions, which have helped us with valuable insights to further improve this work. In response, we conducted additional experiments and carefully revised the manuscript. We believe that these revisions have significantly enhanced the overall quality and clarity of our work.

Specific Comments:

1. The literature cited throughout the manuscript is largely outdated, with many references dating back five years or more. Incorporating recent studies would strengthen the relevance and positioning of the work.

=> Thank you for this helpful suggestion. We have added recent references to better reflect advances in the field and to clarify the positioning of our work (Refs. 71–75). The revised manuscript now cites 41 papers published since 2020.

2. The reported design largely revolves around nanoparticle doping of titanate nanosheets, but does not convincingly reflect the claimed modularity or novel functionality. Furthermore, the manuscript does not sufficiently address existing material challenges or clarify how this system overcomes them.

=> We appreciate this insightful comment. First, we note the general difficulty of constructing photonic crystals using colloidal nanosheets. To form photonic nanostructures, colloidal nanosheets (thickness = ~1 nm) must self-assemble in water to achieve a uniform periodicity of several hundred nanometers. Accordingly, the nanosheets must meet the following requirements: uniform thickness and a high aspect ratio to ensure proper structural ordering, a large surface charge density for strong electrostatic repulsion, and robust structural and colloidal stability for solution processing. As a result, inorganic nanosheets available for

constructing photonic crystals have been restricted to specific compositions, such as titanate [Refs. 19–22], graphene oxide [Refs. 23–32], antimony phosphate [Refs. 33–35], zirconium phosphate [Refs. 36–39], titanium phosphate [Ref. 40], niobate [Refs. 41,42], and clay minerals [Refs. 43–46]. If functional nanosheets could be harnessed to construct photonic crystals, they would offer a new platform for integrating additional functionalities, thereby enabling the development of multi-functional photonic crystals. However, both pre-functionalization (i.e., introducing desired functions into precursor layered crystals prior to their exfoliation into nanosheets) and post-functionalization (i.e., surface modification with functional molecules or nanoparticles after exfoliation) have typically failed to meet the above criteria, and therefore, their use in constructing photonic crystals has yet to be realized. In this context, the key challenge is to develop a universal strategy for synthesizing inorganic nanosheets that not only satisfy the above criteria but also possess additional functionalities for constructing multi-functional photonic crystals.

To overcome this challenge, in this work, we propose a general method to post-functionalize the base nanosheets known to form photonic crystals, while retaining their photonic ability, through surface modification with functional nanoparticles under optimized conditions. Importantly, this approach allows for the modular integration of diverse functionalities into nanosheets simply by varying the nanoparticles, thereby enabling the creation of multi-functional photonic crystals with tunable optical properties. Indeed, using this modular approach, we succeeded in synthesizing functional nanosheets and constructing multi-functional photonic crystals with modularly integrated structural color, plasmonic absorption, and fluorescence. The resultant multi-functional photonic crystals exhibited diverse functionalities, such as (i) photo-responsive structural colors, (ii) magneto-responsive color switching between reflection-based structural colors and absorption-based plasmonic colors, (iii) photonic inks with unique reflection- and absorption-based coloration, and (iv) direct 3D visualization of individual nanosheets within the photonic nanostructure using CLSM, as explained in detail in our response to the reviewer's next comment.

To add this information, we have carefully revised the Introduction (page 4, line 5–12).

3. Although the multifunctionality of the composite material is emphasized, the demonstrated responses to external stimuli remain relatively simple and limited. Given the extensive number of similar reports in recent literature, the current results do not convincingly showcase a significant advancement in functional material design.

=> We appreciate this valuable comment. To further demonstrate the multi-functionality and

versatility of our photonic crystals, we performed additional experiments (Supplementary Figs. 15, 16, 19 and 20 and Supplementary Movie 2 and 3). The unique functionalities arising from the modular integration and intrinsic two-dimensionality of the nanosheets in the photonic crystals are summarized below.

(i) **Photo-responsive structural colors:** The plasmonic nanosheets (AuNP-TiNSs) inherit the efficient photo-thermal conversion capability of the constituent AuNPs on their surface, enabling photo-induced reversible modulation of the structural colors (Fig. 5c).

(ii) **Magneto-responsive color switching:** The hybrid nanosheets inherit the magnetic responsiveness of the pristine titanate nanosheets, allowing for magneto-induced reversible color switching between reflection-based structural colors and absorption-based plasmonic colors (Supplementary Fig. 19).

(iii) **Photonic inks with unique coloration:** Owing to the inherited fluidity and photonic properties, together with the plasmonic properties of the constituent AuNPs or AuNRs, the multi-functional photonic crystals can be used as photonic inks exhibiting combined reflection- and absorption-based coloration (Supplementary Fig. 20).

(iv) **Direct 3D visualization of individual nanosheets:** The fluorescent nanosheets (FSNP-TiNSs) inherit both the efficient fluorescence properties of the constituent FSNPs and the large periodicity of pristine titanate nanosheets, enabling direct 3D visualization of individual nanosheets within the photonic nanostructure using CLSM. This capability allowed us to observe the time-dependent structural relaxation from magnetically oriented to random states (Supplementary Fig. 15 and Supplementary Movie 2) and the dynamic behavior of the nanosheets within giant vesicles (Supplementary Fig. 16 and Supplementary Movie 3). This approach offers a significant advantage for the analysis of the self-assembled nanostructures of colloidal nanosheets, overcoming the limitations of conventional SEM methods that typically require complicated fixation and drying procedures which often disrupt the original architectures.

Overall, these demonstrations strengthen the versatility of our multi-functional photonic crystals, representing a significant advancement beyond previously reported nanosheet-based photonic systems. Accordingly, we have added the information to the main text and the Supplementary Information (Supplementary Figs. 15, 16, 19, and 20 and Supplementary Movie 2 and 3).

Supplementary Fig. 15 | Time-dependent CLSM image of the magnetically treated photonic crystal of FSNP-TiNSs.

Time-dependent confocal laser scanning microscopy (CLSM) images of the magnetically treated photonic crystal of FSNP-TiNSs (0.40 wt%) in a dispersion state using a 550-nm laser.

Supplementary Fig. 16 | Time-dependent CLSM image of the photonic crystal of FSNP-TiNSs within giant vesicles.

Time-dependent confocal laser scanning microscopy (CLSM) images of the photonic crystal of FSNP-TiNSs (0.40 wt%) within giant vesicles using a 460-nm laser for the vesicles and a 550-nm laser for the nanosheets.

Supplementary Fig. 19 | Magneto-responsive structural colors of the multi-functional photonic crystals.

(i) Schematic illustrations, (ii) optical images, and (iii) extinction spectra of the photonic crystals of AuNP-TiNSs (0.50 and 0.60 wt%) and AuNR-TiNSs (0.40 and 0.50 wt%) by changing the direction of the applied magnetic field from the z -axis (left) to the y -axis (right).

Supplementary Fig. 20 | Photonic inks from the multi-functional photonic crystals.

Optical images of the photonic crystals of AuNP-TiNSs and AuNR-TiNSs (0.40, 0.50, and 0.60 wt%) and demonstration as photonic inks with reflection- and absorption-based coloration.

Point-by-point Response to Reviewer #2:

This article focuses on the development of multi-functional 2-D nanosheets and their assembly into 1D liquid photonic crystals (LPCs). The study of these not only deepens our fundamental understanding of anisotropic particle interactions but also enables the rational design of functional materials with tailored optical, mechanical, or electronic properties. In this paper, TiNS colloidal nanosheets functionalized with gold nanoparticles, gold nanorods, or fluorescent silica nanoparticles were prepared for the self-assembly of multifunctional liquid photonic crystals. The structural color features of liquid photonic crystals are tunable by aligning nanosheets with a magnetic field or through light-induced interlayer spacing changes. Also, the integration of fluorescent particles in the system allows for 3D visualization of individual sheets or the self-assembly processes of photonic structures via confocal microscopy. The study addresses an interesting topic, but several key aspects require clarification or further development:

=> We sincerely appreciate your positive assessment that our study addresses an interesting topic. We are also grateful for your constructive comments and suggestions. In response, we conducted additional experiments and carefully addressed your concerns, as detailed below. Accordingly, we have thoroughly revised the manuscript, and we believe that these revisions have significantly improved the overall quality and clarity of our work.

1. As the Bragg equation for 1D PCs ($\lambda = 2nd \cos \theta$) is simple, the authors attribute the observed color changes to variations in interlayer spacing d , and estimate the d according to this formula as well. In their earlier publications, the possible TiNS interlayer distance was also estimated using DLVO theory (Nat. Commun. 7, 12559 (2016)). Since the SAXS experiments have already been carried out, and the resolution of SPring-8 facility is extremely high, it is confusing why they did not provide direct measurement results of interlayer spacing d , through wavevector transfer $q = 4\pi \sin \theta / \lambda$. In addition, by comparing the reflection spectra with SAXS data $q = 4\pi \sin \theta / \lambda$ (see Adv. Mater. 2010, 22, 2871–2880), the effective refractive index n in the Bragg equation ($\lambda = 2nd \cos \theta$) can be calculated, which may provide insight into the electrostatic repulsion between 2D colloids both in the vertical (inter-plane) and the horizontal (in-plane) direction for ordering, or providing practical packing density of the 2D nanosheets within crystalline LPC domains (which should be higher than in the bulk solution).

=> We appreciate this important comment and suggestion. First, we note that the SAXS measurements at the SPring-8 facility (Fig. 2) were designed to evaluate the magnetic responsiveness of the hybrid nanosheets, and therefore, we used low nanosheet

concentrations (0.050 wt%). Under these dilute conditions, the dispersions did not exhibit structural color, and the expected interlayer spacing was too large to be detected within the measurable range even at the SPring-8 facility. Following the reviewer's suggestion, we performed additional SAXS and reflection measurements using the photonic crystal of AuNP-TiNSs (2.0 wt%) that exhibits structural color (Supplementary Fig. 12). The 1D SAXS profile gave an interlayer spacing d of 126 nm. Additionally, using the Bragg equation, we obtained an effective refractive index n of 1.3, which is almost identical to that of water ($n = 1.3$). Although this is reasonable considering the relatively low nanosheet concentration, a further discussion of the nanosheet packing structure is difficult based solely on this result. Accordingly, we have added the information to the main text (page 9, line 8–10) with citing the suggested reference (Ref. 67) and the Supplementary Information (Supplementary Fig. 12).

Supplementary Fig. 12 | SAXS and reflection of the photonic crystal of AuNP-TiNSs.

a,b, (a) 1D small-angle X-ray scattering (SAXS) profile and (b) reflection spectrum of the photonic crystal of AuNP-TiNSs (2.0 wt%).

2. Fluorescent nanoparticles are incorporated into the nanosheets, enabling visualization of the dynamic self-assembly of 2d nanosheets. But the current implementation is limited to static 3D visualization of the photonic nanostructure, which does not fully demonstrate the advantages and advancements offered by this technique. Given the successful incorporation of fluorescent nanoparticles or dyes into the nanosheets and their three-dimensional visualization in previous studies, it is recommended to visualize the dynamic colloidal assembly process, such as nucleation and growth of photonic crystal domains; dynamic phase transition from a disordered to ordered liquid crystalline states, or kinetics of alignment under external fields (magnetic, electric, shear), making the findings more impactful.

=> We appreciate this insightful suggestion. In response, we performed time-dependent CLSM observations to visualize the dynamic behavior of the magnetically oriented nanosheets in their photonic crystals. We successfully visualized the structural relaxation of the nanosheets from a magnetically oriented state to a random state (Supplementary Fig. 15 and

Supplementary Movie 2). Furthermore, we succeeded in visualizing the movements of the nanosheets even within giant vesicles (Supplementary Fig. 16 and Supplementary Movie 3), further demonstrating the versatility of the present technique. Accordingly, we have added the information to the main text (page 11, line 14–17) and the Supplementary Information (Supplementary Figs. 15 and 16 and Supplementary Movies 2 and 3).

Supplementary Fig. 15 | Time-dependent CLSM image of the magnetically treated photonic crystal of FSNP-TiNSs.

Time-dependent confocal laser scanning microscopy (CLSM) images of the magnetically treated photonic crystal of FSNP-TiNSs (0.40 wt%) in a dispersion state using a 550-nm laser.

Supplementary Fig. 16 | Time-dependent CLSM image of the photonic crystal of FSNP-TiNSs within giant vesicles.

Time-dependent confocal laser scanning microscopy (CLSM) images of the photonic crystal of FSNP-TiNSs (0.40 wt%) within giant vesicles using a 460-nm laser for the vesicles and a 550-nm laser for the nanosheets.

3. It is recommended that the authors present absolute reflectance measurement results instead of relative reflectance of 1D LPCs, and their transmittance information is also necessary for understanding of their optical properties.

=> In response to this constructive comment, we conducted additional experiments using UV-Vis spectroscopy in both transmission and reflection modes and provided the absolute reflectance, transmittance, and extinction spectra of the multi-functional photonic crystals (Supplementary Fig. 13), which contribute to the understanding of their optical properties. In the main manuscript, we primarily presented extinction spectra obtained in transmission

mode, since it is necessary to simultaneously evaluate the structural colors of photonic crystals (i.e., reflection) as well as plasmonic colors (i.e., absorption). Accordingly, we have added the information to the main text (page 9, line 19–23) and the Supplementary Information (Supplementary Fig. 13).

Supplementary Fig. 13 | Optical properties of the photonic crystal of AuNP-TiNSs.

a–d, (i) Extinction and (ii) transmittance spectra measured in transmission mode and (iii) reflectance spectra measured in reflection mode of the photonic crystal of pristine TiNSs (**a**: 0.40 wt%; **c**: 0.50 wt%), (**b**) the photonic crystal of AuNP-TiNSs (0.40 wt%), and (**d**) the photonic crystal of AuNR-TiNSs (0.50 wt%) in 1-mm-thick quartz cuvettes ($40 \times 10 \times 1$ mm).

4. The influence of ion strength, PH changes on the reflectance spectra of 1D LPCs, and long-term stability of the systems (since the Au and fluorescent parties are static electricity adsorbed) should be investigated.

⇒ In response to these valuable suggestions, we evaluated the influence of ionic strength and pH on the structural colors of the multi-functional photonic crystals. We found that increasing ionic strength from 0 to 0.20 mM and decreasing the pH from 6.92 to 6.36 induced a blue shift of the structural colors due to reduced electrostatic repulsion between the nanosheets

(Supplementary Fig. 17). Additionally, we investigated the long-term stability of both the hybrid nanosheets and their photonic crystals, confirming that the plasmonic and fluorescent properties of the hybrid nanosheets (Supplementary Figs. 4–6) as well as the photonic properties (Supplementary Fig. 14) were preserved for 7 days. Accordingly, we have added the information to the main text (page 7, line 3–8; page 10, line 1–2; page 12, line 18–21) and the Supplementary Information (Supplementary Figs. 4–6, 14, and 17).

Supplementary Fig. 4 | Long-term and thermal stability of AuNP-TiNS.

a–c, Polarized optical microscopy (POM) images under crossed Nicols (upper) and extinction spectra (lower) of aqueous dispersions (0.20 wt%) of (a) as-synthesized AuNP-TiNSs, (b) AuNP-TiNSs after 7 days of storage, and (c) AuNP-TiNSs after heating at 70 °C for 30 min in 1-mm-thick quartz cuvettes ($40 \times 10 \times 1$ mm).

Supplementary Fig. 5 | Long-term and thermal stability of AuNR-TiNS.

a–c, Polarized optical microscopy (POM) images under crossed Nicols (upper) and extinction spectra (lower) of aqueous dispersions (0.20 wt%) of (a) as-synthesized AuNR-TiNSs, (b) AuNR-TiNSs after 7 days of storage, and (c) AuNR-TiNSs after heating at 70 °C for 30 min in 1-mm-thick quartz cuvettes ($40 \times 10 \times 1$ mm).

Supplementary Fig. 6 | Long-term and thermal stability of FSNP-TiNS.

a–c, Polarized optical microscopy (POM) images under crossed Nicols (upper) and fluorescence spectra (lower) of aqueous dispersions (0.20 wt%) of **(a)** as-synthesized FSNP-TiNSs, **(b)** FSNP-TiNSs after 7 days of storage, and **(c)** FSNP-TiNSs after heating at 70 °C for 30 min in 1-mm-thick quartz cuvettes ($40 \times 10 \times 1$ mm).

Supplementary Fig. 14 | Long-term stability of the photonic crystal of AuNP-TiNSs.

Extinction spectra of the photonic crystal of AuNP-TiNSs (0.40 wt%) in 1-mm-thick quartz cuvettes ($40 \times 10 \times 1$ mm) before and after 7 days of storage.

Supplementary Fig. 17 | Structural colors of the photonic crystal of AuNP-TiNSs as a function of ionic concentration and pH.

a,b, Optical images (upper) and extinction spectra (lower) of the magnetically treated photonic crystals of AuNP-TiNSs (0.50 wt%) in 1-mm-thick quartz cuvettes ($40 \times 10 \times 1$ mm) at **(a)** ionic concentrations ($[\text{NaCl}] = 0, 0.05, 0.10, 0.15,$ and 0.20 mM) and **(b)** pH (6.92, 6.79, 6.60, 6.44, and 6.36). The pH was adjusted by adding an aqueous HCl solution.

- For self-assembly of spherical, polyhedral, or 2d nanoparticles, the ordered packing of particles are supposed to feature a higher packing density, and thereby a stronger reflectance and larger wavelengths of reflective peaks, when compared with disordered systems. But the peak positions of Fig. 4a and 4b showed the opposite tendency, which should be caused by external magnetic field forces, thereby theoretical fundamentals or experimental results of collective effects of magnetic field and electrostatic repulsion on the self-assembly and optical properties of 1D LPCs should be provided.

=> We appreciate this important comment. According to the Derjaguin-Landau-Verwey-Overbeek (DLVO) theory, colloidal nanosheets in aqueous dispersion are likely trapped in the secondary minimum of their interaction potential, resulting from the balance between electrostatic repulsion and van der Waals attraction. However, just after the preparation of photonic crystals, the movements of the nanosheets are restricted by neighboring nanosheets due to their high aspect ratio and relatively high concentrations, thereby kinetically trapping them at an interlayer spacing larger than the most favorable one. In contrast, the magnetic treatment induces uniaxial orientation of the nanosheets, allowing for their 1D displacement along their normal direction to achieve their most favorable interlayer spacing. Since the

structural colors are directly determined by the interlayer spacing according to Bragg's law, the reduction of the spacing after the magnetic treatment explains the blue shift of the structural colors observed in Fig. 4a and 4b.

Point-by-point Response to Reviewer #3:

This manuscript titled "Multi-functional photonic crystals of modular nanosheets" presents a significant advancement in the field of photonic crystals. The authors introduce a novel and universal modular strategy to synthesize functional hybrid nanosheets and construct multi-functional photonic crystals with customizable properties through self-assembly. The work is highly innovative as it successfully integrates multiple functionalities into nanosheet-based photonic crystals, which has been a considerable challenge due to the strict structural and colloidal requirements. The strategy not only leverages the unique advantages of colloidal nanosheets but also demonstrates impressive controllability and versatility in optical functionalities, such as structural color, plasmonic absorption, and fluorescence. The logical flow of the manuscript is clear, starting from the synthesis of hybrid nanosheets to the construction and characterization of photonic crystals, providing a comprehensive understanding of the research. I believe that with a minor revision addressing the following concerns, it will be more robust and convincing.

=> We sincerely appreciate your positive evaluation of our work as well as your valuable comments and suggestions. In response, we conducted additional experiments and carefully addressed your concerns, as detailed below. Accordingly, we have thoroughly revised the manuscript, and we believe that these revisions have significantly improved the overall quality and clarity of our work.

1. Regarding the adsorption of positively charged nanoparticles on negatively charged sheets, the authors should investigate how the amount of added nanoparticles affects the formation of composite structures and their properties. They mention that adding too much can affect dispersibility but fail to provide supporting data.

=> We sincerely appreciate these valuable suggestions. In response, we investigated how the amount of added nanoparticles affects the formation behavior and optical properties of hybrid nanosheets (Supplementary Fig. 8). We observed that an excessive amount of positively charged nanoparticles (AuNPs) led to aggregation of the nanosheets in optical images and disappearance of birefringence in polarized optical images, although the plasmonic properties of AuNPs were preserved. In contrast, the addition of a smaller amount of AuNPs resulted in colloiddally stable hybrid nanosheets maintaining the liquid-crystalline behavior, but they exhibited almost no plasmonic properties. These results highlight the importance of optimizing the concentration of added nanoparticles. We also provided data showing that excessive addition of other types of nanoparticles (AuNRs and FSNPs) similarly caused aggregation (Supplementary Fig. 7). Accordingly, we have added the information to the main

text (page 7, line 9–14) and the Supplementary Information (Supplementary Figs. 7 and 8).

Supplementary Fig. 7 | TiNS dispersions after addition of excessive nanoparticles.

a–c, Optical microscopy images (upper) and polarized optical microscopy (POM) images under crossed Nicols (lower) of aqueous dispersions of TiNSs (0.20 wt%) in 1-mm-thick quartz cuvettes ($40 \times 10 \times 1$ mm) after adding nanoparticles at 10 \times their respective optimized amounts (**a**: AuNPs; **b**: AuNRs; **c**: FSNPs).

Supplementary Fig. 8 | TiNS dispersions after addition of varying amounts of AuNPs.

Optical microscopy images (upper), polarized optical microscopy (POM) images under crossed Nicols (middle), and normalized extinction spectra (lower) of aqueous dispersions of TiNSs (0.20 wt%) in 1-mm-thick quartz cuvettes ($40 \times 10 \times 1$ mm) after adding AuNPs at 0.1 \times , 0.5 \times , 1 \times , 5 \times , or 10 \times the optimized amount.

Additionally, the authors need to clarify why maintaining a negative charge is essential for stability and whether a positive charge could achieve the same effect through electrostatic repulsion.

=> According to the Derjaguin-Landau-Verwey-Overbeek (DLVO) theory, the colloidal stability of nanosheets in aqueous dispersions is generally governed by electrostatic repulsion, and therefore, maintaining a high surface charge is crucial to avoid aggregation of nanosheets, regardless of whether the charge is negative or positive. Consequently, as the reviewer pointed out, we also expect that our strategy could, in principle, be applied to positively charged nanosheets combined with negatively charged nanoparticles, although most reported nanosheets are negatively charged.

2. The authors emphasize the importance of charge interactions for maintaining photonic crystals. For multi-component systems, it is crucial to consider the local heterogeneity and double-electricity resulting from the adsorption of positively charged particles on negatively charged surfaces, which significantly influence the overall properties. The authors are advised to conduct additional control experiments and simulations to gain deeper insights into the role of charge interactions.

=> In response to this insightful suggestion, we conducted additional control experiments to compare the photonic properties of pristine titanate nanosheets and hybrid nanosheets using UV-Vis spectroscopy in both transmission and reflection modes (Supplementary Fig. 13). We found that the photonic peak positions are almost identical, although the peaks of hybrid nanosheets are broader than those of pristine titanate nanosheets. These results suggest that the average charge interactions are comparable, while local heterogeneity induced by the adsorption of positively charged nanoparticles may cause slight structural disorder in the photonic nanostructures, as the reviewer noted. Accordingly, we have added the information to the main text (page 9, line 19–23) and the Supplementary Information (Supplementary Fig. 13).

Supplementary Fig. 13 | Optical properties of the photonic crystal of AuNP-TiNSs.

a–d, (i) Extinction and (ii) transmittance spectra measured in transmission mode and (iii) reflectance spectra measured in reflection mode of the photonic crystal of pristine TiNSs (**a**: 0.40 wt%; **c**: 0.50 wt%), (**b**) the photonic crystal of AuNP-TiNSs (0.40 wt%), and (**d**) the photonic crystal of AuNR-TiNSs (0.50 wt%) in 1-mm-thick quartz cuvettes ($40 \times 10 \times 1$ mm).

3. The practical applications of the synthesized photonic crystals should be demonstrated. While the crystals exhibit excellent controllability and multi-functionality, the authors do not show their performance in real-world applications. For example, could the rotation of nanosheets be observed within cells?

=> In response to this constructive suggestion, we conducted two demonstrations. (i) Following the reviewer's suggestion, we incorporated our photonic crystals composed of fluorescent nanosheets into fluorescently labeled cell-like giant vesicles and performed time-dependent CLSM observations. Interestingly, we observed the movement of the nanosheets even within the vesicles (Supplementary Fig. 16 and Supplementary Movie 3). (ii) Additionally, we demonstrated the versatility of our multi-functional photonic crystals as photonic inks exhibiting complex colors arising from the combination of photonic reflection and plasmonic absorption (Supplementary Fig. 20). Accordingly, we have added the information to the main

text (page 11, line 16–17; page 13, line 16–19) and the Supplementary Information (Supplementary Figs. 16 and 20 and Supplementary Movie 3).

Supplementary Fig. 16 | Time-dependent CLSM image of the photonic crystal of FSNP-TiNSs within giant vesicles.

Time-dependent confocal laser scanning microscopy (CLSM) images of the photonic crystal of FSNP-TiNSs (0.40 wt%) within giant vesicles using a 460-nm laser for the vesicles and a 550-nm laser for the nanosheets.

Supplementary Fig. 20 | Photonic inks from the multi-functional photonic crystals.

Optical images of the photonic crystals of AuNP-TiNSs and AuNR-TiNSs (0.40, 0.50, and 0.60 wt%) and demonstration as photonic inks with reflection- and absorption-based coloration.

4. The process of adsorbing nanoparticles onto nanosheets is not novel in itself. The innovation lies in maintaining the characteristics of photonic crystals while forming composite structures. The authors should further discuss this aspect in the Introduction to highlight the novelty of their approach, these discussions can also make the article accessible to a wider range of readers.

=> We sincerely appreciate the constructive suggestion. In response, we have carefully revised the Introduction (page 4, line 5–12).

Point-by-point Response to Reviewer #4:

This work proposes an innovative modular strategy to construct multifunctional photonic crystals with structural color, plasma absorption, and fluorescence by combining functional nanoparticles (AuNPs, AuNRs, FSNPs) with titanate nanosheets (TiNSs) through electrostatic attraction. The multifunctional photonic crystals exhibit good color and structural tunability by applying external stimuli (light, magnetic fields, etc.) and may find new applications in optical devices. Overall, this work is interesting and well organized, and therefore I would recommend the publication of this work after proper revision. Here are some comments.

=> We sincerely appreciate your positive evaluation of our work as well as your insightful comments and suggestions. In response, we conducted additional experiments and carefully revised the manuscript. Accordingly, we believe that these revisions have significantly improved the overall quality and clarity of our work.

1. How is the optimal concentration of functional nanoparticles and nanosheets determined? Do variations in these concentrations influence the optical properties of the resulting hybrid nanosheets?

=> We sincerely appreciate these important questions. The optimal concentration was determined by gradually adjusting the ratio between the functional nanoparticles and the nanosheets. For example, when an excessive amount of positively charged AuNPs was added to the nanosheet dispersion, aggregation of the nanosheets and loss of their liquid-crystalline behavior were observed, although the plasmonic properties of AuNPs were still preserved (Supplementary Fig. 8). In contrast, the addition of a smaller amount of AuNPs resulted in colloiddally stable hybrid nanosheets that maintained the liquid-crystalline behavior, but they exhibited almost no plasmonic properties. Accordingly, we selected the ratio as the optimal point at which the hybrid nanosheets maintain their liquid-crystalline behavior without aggregation while retaining sufficient plasmonic properties. As the reviewer pointed out, the optical properties of the hybrid nanosheets are indeed influenced by the ratio between the functional nanoparticles and nanosheets, as confirmed by extinction spectra and polarized optical microscopy images. Accordingly, we have added the information to the main text (page 7, line 9–14) and the Supplementary Information (Supplementary Fig. 8).

Supplementary Fig. 8 | TiNS dispersions after addition of varying amounts of AuNPs.

Optical microscopy images (upper), polarized optical microscopy (POM) images under crossed Nicols (middle), and normalized extinction spectra (lower) of aqueous dispersions of TiNSs (0.20 wt%) in 1-mm-thick quartz cuvettes ($40 \times 10 \times 1$ mm) after adding AuNPs at 0.1 \times , 0.5 \times , 1 \times , 5 \times , or 10 \times the optimized amount.

2. Is there competitive adsorption between AuNRs and FS NPs on TiNSs surfaces? Is the particle distribution across nanosheets uniform? If not, how does adsorption heterogeneity affect optical performance?

=> To evaluate competitive adsorption, we varied the amount of FS NPs added to the nanosheet dispersion while keeping the amount of added AuNRs constant and found that the plasmonic peaks of the hybrid nanosheets (AuNR/FSNP-TiNSs) remained essentially unchanged (Supplementary Fig. 9). This result indicates that there is no significant competitive adsorption between AuNRs and FS NPs and that all added nanoparticles are effectively attached to the TiNS surfaces, likely because the nanoparticle concentration is much lower than the adsorption capacity of the nanosheets. TEM images of AuNR/FSNP-TiNSs show some local heterogeneity in nanoparticle attachment within individual nanosheets, but the overall attachment pattern is generally uniform across different nanosheets. Consistently, the optical properties of AuNR/FSNP-TiNSs are inherited from the constituent nanoparticles (AuNRs and FS NPs) and are almost identical to those of nanoparticles alone, as confirmed by the extinction and fluorescence spectra in Fig. 2. Accordingly, we have added the information to the main text (page 7, line 18–21) and the Supplementary Information

(Supplementary Fig. 9).

Supplementary Fig. 9 | TiNS dispersions after addition of AuNRs and FS NPs at varying ratios.

a–c, Extinction spectra of aqueous dispersions of TiNSs (0.20 wt%) in 1-mm-thick quartz cuvettes ($40 \times 10 \times 1$ mm) after adding AuNRs and FS NPs at different multiples of their respective optimized amounts (a: AuNRs 1 \times and FS NPs 0 \times ; b: AuNRs 1 \times and FS NPs 0.5 \times ; c: AuNRs 1 \times and FS NPs 1 \times).

3. Quantitative spectral data after a long-term storage and after heating are suggested to be supplemented to further prove the stability of the prepared nanosheets.

=> In response to this constructive suggestion, we measured the extinction and fluorescence spectra of the hybrid nanosheets (AuNP-TiNSs, AuNR-TiNSs, and FSNP-TiNSs) after 7 days of storage and after heating at 70 °C for 30 min (Supplementary Figs. 4–6). The results confirmed that their optical properties remained essentially unchanged, further supporting their long-term and thermal stability. Accordingly, we have added the information to the main text (page 7, line 3–8) and the Supplementary Information (Supplementary Figs. 4–6).

Supplementary Fig. 4 | Long-term and thermal stability of AuNP-TiNS.

a–c, Polarized optical microscopy (POM) images under crossed Nicols (upper) and extinction spectra (lower) of

aqueous dispersions (0.20 wt%) of (a) as-synthesized AuNP-TiNSs, (b) AuNP-TiNSs after 7 days of storage, and (c) AuNP-TiNSs after heating at 70 °C for 30 min in 1-mm-thick quartz cuvettes (40 × 10 × 1 mm).

Supplementary Fig. 5 | Long-term and thermal stability of AuNR-TiNS.

a–c, Polarized optical microscopy (POM) images under crossed Nicols (upper) and extinction spectra (lower) of aqueous dispersions (0.20 wt%) of (a) as-synthesized AuNR-TiNSs, (b) AuNR-TiNSs after 7 days of storage, and (c) AuNR-TiNSs after heating at 70 °C for 30 min in 1-mm-thick quartz cuvettes (40 × 10 × 1 mm).

Supplementary Fig. 6 | Long-term and thermal stability of FSNP-TiNS.

a–c, Polarized optical microscopy (POM) images under crossed Nicols (upper) and fluorescence spectra (lower) of aqueous dispersions (0.20 wt%) of (a) as-synthesized FSNP-TiNSs, (b) FSNP-TiNSs after 7 days of storage, and (c) FSNP-TiNSs after heating at 70 °C for 30 min in 1-mm-thick quartz cuvettes (40 × 10 × 1 mm).

4. The explanation of the magnetic field switching structure color is insufficient. The change of the AuNP-TiNSs spectrum with the change of the magnetic field direction during the process of magnetic field regulating the structure color should be supplemented to prove that the disappearance of color satisfies the Bragg diffraction equation.

=> In response to this insightful comment, we measured the UV-Vis spectra of AuNP-TiNS photonic crystals by changing the angle of the applied magnetic field relative to the surface of the quartz cuvette from 90° to 40° (Supplementary Fig. 18). The results confirm that the peak shift of the structural color is consistent with Bragg's law. Accordingly, we have added the information to the main text (page 13, line 10–12) and the Supplementary Information (Supplementary Fig. 18).

Supplementary Fig. 18 | Structural colors of the photonic crystal of AuNP-TiNSs as a function of magnetic field angle.

a,b, (a) extinction spectra and (b) optical images of the magnetically treated photonic crystals of AuNP-TiNSs (0.50 wt%) in 1-mm-thick quartz cuvettes (40 × 10 × 1 mm) by varying the angle of the applied magnetic field with respect to the cuvette surface. **c**, Observed wavelength of the first-order structural colors and calculated values from Bragg's law as a function of the magnetic field angle.

5. Why does AuNR/FSNP-TiNSs exhibit a lower zeta potential than pure TiNSs?

=> We sincerely appreciate this important question. As the reviewer noted, the zeta potential of AuNR/FSNP-TiNSs is slightly lower than that of pure TiNSs. However, this difference is minor compared with the other hybrid nanosheets (AuNP-TiNSs, AuNR-TiNSs, and FSNP-TiNSs), and we consider, given the low concentration of added nanoparticles, that the overall surface charge of these hybrid nanosheets are essentially comparable to those of pure TiNSs.

Indeed, the UV-Vis spectra show that these hybrid nanosheets form photonic nanostructures exhibiting photonic properties similar to those of pure TiNS photonic crystals (Supplementary Fig. 13), further supporting the comparable charge interactions.

6. What are the advantages of using nanosheet compared to nanospheres (Adv. Funct. Mater. 2024, 2411670 and Nature Communications, 2024, 15, 5643) in constructing multifunctional photonic crystals? These works should be discussed in the revision.

=> We sincerely appreciate this constructive suggestion. Compared with nanospheres, the nanosheets used in this work offer distinct advantages arising from their intrinsic two-dimensionality and modular integration capability, which enable the construction of multifunctional photonic crystals with unique anisotropic tunability and modular optical properties. Their high aspect ratio and large surface area facilitate the formation of 1D photonic nanostructures and the effective integration of additional functionalities, which are difficult to achieve with isotropic nanospheres. To further demonstrate the versatility of our photonic crystals, we performed additional experiments (Supplementary Figs. 15, 16, 19, and 20 and Supplementary Movie 2 and 3). The unique functionalities arising from the intrinsic two-dimensionality and modular integration capability of the nanosheets in the photonic crystals are summarized below.

(i) **Photo-responsive structural colors:** The plasmonic nanosheets (AuNP-TiNSs) inherit the efficient photo-thermal conversion capability of the constituent AuNPs on their surface, enabling photo-induced reversible modulation of the structural colors (Fig. 5c).

(ii) **Magneto-responsive color switching:** The hybrid nanosheets inherit the magnetic responsiveness of the pristine titanate nanosheets, allowing for magneto-induced reversible color switching between reflection-based structural colors and absorption-based plasmonic colors (Supplementary Fig. 19).

(iii) **Photonic inks with unique coloration:** Owing to the inherited fluidity and photonic properties, together with the plasmonic properties of the constituent AuNPs or AuNRs, the multi-functional photonic crystals can be used as photonic inks exhibiting combined reflection- and absorption-based coloration (Supplementary Fig. 20).

(iv) **Direct 3D visualization of individual nanosheets:** The fluorescent nanosheets (FSNP-TiNSs) inherit both the efficient fluorescence properties of the constituent FSNPs and the large periodicity of pristine titanate nanosheets, enabling direct 3D visualization of individual nanosheets within the photonic nanostructure using CLSM. This capability

allowed us to observe the time-dependent structural relaxation from magnetically oriented to random states (Supplementary Fig. 15 and Supplementary Movie 2) and the dynamic behavior of the nanosheets within giant vesicles (Supplementary Fig. 16 and Supplementary Movie 3). This approach offers a significant advantage for the analysis of the self-assembled nanostructures of colloidal nanosheets, overcoming the limitations of conventional SEM methods that typically require complicated fixation and drying procedures which often disrupt the original architectures.

Overall, these demonstrations strengthen the versatility of our multi-functional photonic crystals, representing a significant advancement beyond previously reported nanosheet-based photonic systems. Accordingly, we have added the information to the main text (page 15, line 5–6) with citing suggested references (Refs. 71–75) and the Supplementary Information (Supplementary Figs. 15, 16, 19 and 20 and Supplementary Movie 2 and 3).

Supplementary Fig. 15 | Time-dependent CLSM image of the magnetically treated photonic crystal of FSNP-TiNSs.

Time-dependent confocal laser scanning microscopy (CLSM) images of the magnetically treated photonic crystal of FSNP-TiNSs (0.40 wt%) in a dispersion state using a 550-nm laser.

Supplementary Fig. 16 | Time-dependent CLSM image of the photonic crystal of FSNP-TiNSs within giant vesicles.

Time-dependent confocal laser scanning microscopy (CLSM) images of the photonic crystal of FSNP-TiNSs (0.40 wt%) within giant vesicles using a 460-nm laser for the vesicles and a 550-nm laser for the nanosheets.

Supplementary Fig. 19 | Magneto-responsive structural colors of the multi-functional photonic crystals.

(i) Schematic illustrations, (ii) optical images, and (iii) extinction spectra of the photonic crystals of AuNP-TiNSs (0.50 and 0.60 wt%) and AuNR-TiNSs (0.40 and 0.50 wt%) by changing the direction of the applied magnetic field from the z-axis (left) to the y-axis (right).

Supplementary Fig. 20 | Photonic inks from the multi-functional photonic crystals.

Optical images of the photonic crystals of AuNP-TiNSs and AuNR-TiNSs (0.40, 0.50, and 0.60 wt%) and demonstration as photonic inks with reflection- and absorption-based coloration.

7. In Fig. 5b, when a magnetic field with a vertical viewing Angle is applied, it can be observed that the secondary diffraction peaks overlap with the plasma absorption peaks. Why is the intensity at this time higher than that of the plasma absorption peaks when a magnetic field with a parallel viewing Angle is applied?

=> First, we note that the UV-Vis spectra in Fig. 5b were recorded in transmission mode and are plotted as extinction spectra to evaluate both reflection-based structural color and absorption-based plasmonic properties. In this case, both the reflection of the structural color and plasmonic absorption increase the intensity in the spectra. Therefore, when the second-order peak of structural color overlaps with the plasmonic absorption peak, the peak intensity becomes higher. In contrast, the absence of the photonic contribution leads to a lower peak intensity. To avoid any possible confusion, we now state unambiguously in the main text that Fig. 5b shows extinction spectra measured in transmission mode.

8. Why are the intensities of the first-order diffraction peaks in Fig. 5 all significantly lower than those of the second-order diffraction peaks?

=> We sincerely appreciate this important question. This intensity difference in Fig. 5 can be attributed to the structural asymmetry of our photonic crystals, consisting of alternating ultrathin nanosheet layers (~1 nm) and much thicker water layers (several hundred nanometers). In our previous work, we constructed a simple multilayer model of TiNS photonic crystals and confirmed that the simulation based on the Berreman 4×4 matrix method reproduced the experimental UV-Vis spectrum, in which the first-order peak intensity was lower than that of the second-order peak (see the following Figure).

Experimental and simulated UV-Vis-NIR spectra of the TiNS photonic crystals.

Adapted with permission from Ref. 22. Copyright 2023 Wiley-VCH GmbH.

9. The authors are suggested to provide the reflective spectra of the pure Ti nanosheets-based photonic crystals and multifunctional photonic crystals to show the influence of Au nanorods on the photonic bandgaps.

=> In response to this insightful suggestion, we conducted additional experiments to evaluate the photonic properties of pure titanate nanosheets and hybrid nanosheets using UV-Vis spectroscopy in both transmission and reflection modes (Supplementary Fig. 13). We found that photonic peak positions are almost identical, although the peaks of hybrid nanosheets are broader than those of pure titanate nanosheets. These results suggest that the average charge interactions are comparable, while local heterogeneity within the nanosheets induced by the adsorption of positively charged nanoparticles may cause slight structural disorder in the photonic nanostructures. Accordingly, we have added the information to the main text (page 10, line 15–20) and the Supplementary Information (Supplementary Fig. 13).

Supplementary Fig. 13 | Optical properties of the photonic crystal of AuNP-TiNSs.

a–d, (i) Extinction and (ii) transmittance spectra measured in transmission mode and (iii) reflectance spectra measured in reflection mode of the photonic crystal of pristine TiNSs (**a**: 0.40 wt%; **c**: 0.50 wt%), (**b**) the photonic crystal of AuNP-TiNSs (0.40 wt%), and (**d**) the photonic crystal of AuNR-TiNSs (0.50 wt%) in 1-mm-thick quartz cuvettes ($40 \times 10 \times 1$ mm).

Response to the Reviewers

We sincerely appreciate the reviewers' positive evaluation and recommendation of our revised manuscript for publication in *Nature Communications*. Thanks to the reviewers' constructive comments and suggestions, we believe that the overall quality and clarity of our revised manuscript have been significantly improved. We would like to thank all the reviewers for taking the time and effort to strengthen our manuscript.

Reviewer #2:

The authors have carefully addressed all the comments raised during the first review round. The revised manuscript now presents a more comprehensive study on the modular construction of multifunctional photonic crystals. The additional SAXS measurements, optical characterizations (absolute reflectance and transmittance), and dynamic CLSM visualization experiments significantly strengthen the technical rigor of the work. The newly included discussions on the effects of ionic strength, pH, and long-term stability, as well as the expanded explanation of the charge-interaction mechanism and magnetic/orientation control, effectively resolve the previous concerns. Overall, the manuscript has been substantially improved in both scientific content and presentation quality. The authors' responses are satisfactory and the revisions have enhanced the clarity, completeness, and impact of the work.

I find the manuscript suitable for publication in its current form.

Recommendation: Accept as it stands.

Reviewer #3:

The author has answered all the raised questions, I think its good enough for publication without the need for further revision.

Reviewer #4:

The authors have carefully addressed my issues, and I am satisfied with the revision. Therefore, I recommend the acceptance of this work without further change.